# Dynamic disorganization of synaptic NMDA receptors triggered by autoantibodies from psychotic patients

Julie Jézéquel[1,2], Emily M. Johansson[1,2], Julien P. Dupuis[1,2], Véronique Rogemond[3,4,5], Hélène Gréa[1,2], Blanka Kellermayer[1,2], Nora Hamdani[6,7,8], Emmanuel Le Guen[6,7,8], Corentin Rabu[6,7,8], Marilyn Lepleux[1,2], Marianna Spatola[9], Elodie Mathias[3,4,5], Delphine Bouchet[1,2], Amy J. Ramsey[10], Robert H. Yolken[11], Ryad Tamouza[12,13], Josep Dalmau[9,14], Jérôme Honnorat[3,4,5], Marion Leboyer[6,7,8] & Laurent Groc[1,2]

The identification of circulating autoantibodies against neuronal receptors in neuropsychiatric disorders has fostered new conceptual and clinical frameworks. However, detection reliability, putative presence in different diseases and in health have raised questions about potential pathogenic mechanism mediated by autoantibodies. Using a combination of single molecule-based imaging approaches, we here ascertain the presence of circulating auto-antibodies against glutamate NMDA receptor (NMDAR-Ab) in about 20% of psychotic patients diagnosed with schizophrenia and very few healthy subjects. NMDAR-Ab from patients and healthy subjects do not compete for binding on native receptor. Strikingly, NMDAR-Ab from patients, but not from healthy subjects, specifically alter the surface dynamics and nanoscale organization of synaptic NMDAR and its anchoring partner the EphrinB2 receptor in heterologous cells, cultured neurons and in mouse brain. Functionally, only patients' NMDAR-Ab prevent long-term potentiation at glutamatergic synapses, while leaving NMDAR-mediated calcium influx intact. We unveil that NMDAR-Ab from psychotic patients alter NMDAR synaptic transmission, supporting a pathogenically relevant role.

[1] Univ. de Bordeaux, Interdisciplinary Institute for Neuroscience, UMR 5297, 33077 Bordeaux, France. [2] CNRS, IINS UMR 5297, 33077 Bordeaux, France. [3] Institut NeuroMyoGene INSERM U1217/CNRS, UMR 5310, Lyon 69007, France. [4] Hospices Civils de Lyon, Hôpital Neurologique, 69677 Bron, France. [5] Université de Lyon-Université Claude Bernard Lyon 1, 69008 Lyon, France. [6] University Paris Est Créteil, Psychiatry department, Hopitaux Universitaires Henri Mondor, AP-HP, DHU PePSY, 94010 Créteil, France. [7] Translational Psychiatry Laboratory, INSERM U955, 94010 Créteil, France. [8] FondaMental Foundation, 94010 Créteil, France. [9] Biomedical Research Institute August Pi i Sunyer (IDIBAPS), University of Barcelona, Catalan Institution for Research and Advanced Studies (ICREA), 08036 Barcelona, Spain. [10] Department of Physiology, Department of Pharmacology and Toxicology, University of Toronto, Toronto, ON, Canada M5S 1A8. [11] The Johns Hopkins University School of Medicine, Stanley Division of Developmental Neurovirology, Baltimore, MD 21287, USA. [12] INSERM, U1160, Hôpital Saint Louis, Paris F75010, France. [13] Laboratoire Jean Dausset, LabEx Transplantex, Hôpital Saint Louis, Paris F75010, France. [14] Department of Neurology, University of Pennsylvania, Pennsylvania, PA 19104, USA. Julien P. Dupuis, Véronique Rogemond, Hélène Gréa, and Blanka Kellermayer contributed equally to this work. Jérôme Honnorat, Marion Leboyer and Laurent Groc jointly supervised this work. Correspondence and requests for materials should be addressed to L.G. (email: laurent.groc@u-bordeaux.fr)

O ver the past decade, autoantibodies against neuronal receptors have been increasingly identified in neuropsychiatric disorders[1–6]. These disease-related autoantibodies had generated excitement towards molecular and cellular dissection of psychiatric disorders and has fostered debate on how to identify the patients that may benefit from immunotherapy[7]. Several neurological diseases such as autoimmune encephalitis are indeed well-defined and treated after the identification of autoantibodies against neurotransmitter receptors and ion channels[7, 8]. In the psychiatric field, the link between psychotic disorders, particularly schizophrenia (SCZ), and immune system dysregulations including autoimmunity, is a concept that regained strong support thanks to the better characterization of inflammatory-induced psychotic symptoms and autoimmune encephalitis[9]. The best characterized encephalitis is the anti-N-methyl-D-aspartate receptors (NMDAR)-associated condition in which autoantibodies directed against the GluN1 subunit of NMDAR (NMDAR-Ab) are detected and associated with psychotic symptoms and catatonia, followed by profound neurologic deterioration[10, 11]. In contrast, studies exploring the presence of circulating NMDAR-Ab in SCZ patients have produced contradictory outcomes, with detection prevalence ranging from 0 to < 20%[12–24]. Several reasons for these discrepancies have been proposed and debated, such as different sensitivities and specificities between detection methods[2, 25–27]. From a clinical point-of-view, immunotherapy treatment of patients with acute psychosis and NMDAR-Ab was associated with good outcomes, especially for psychotic patients resistant to anti-psychotics[28, 29]. Although large randomized investigations are surely needed to ascertain NMDAR-Ab pathological role in psychosis, the functional interplay between NMDAR-Ab and psychotic symptoms represents a promising area of biomedical research.

However, the understanding of the mechanism(s) underlying the molecular and behavioral dysfunctions triggered by NMDAR-Ab is still in its infancy. The glutamatergic model of psychosis and SCZ is increasingly accepted as part of the etiopathology based on the discovery that certain NMDAR blockers induce SCZ-like psychosis, reproducing both positive and negative symptoms[30]. The anti-NMDAR encephalitis patients' cerebrospinal fluid (CSF) only slightly prolongs NMDAR open time and purified NMDA-Ab do not alter NMDAR-mediated calcium influx[31, 32], suggesting that NMDAR-Ab-related psychosis is not directly due to receptor blockade. NMDAR-Ab mostly target extracellular epitopes of the GluN1 subunit[31], which is an obligatory subunit that associate with GluN2 and/or GluN3 subunits to form functional tetrameric NMDAR[33]. In addition to the exocytosis/endocytosis cycle, surface NMDAR are constantly trafficked to and from the glutamatergic synapse, in order to ensure the stability of the synaptic pool[34]. NMDAR-Ab from patients with NMDAR encephalitis strongly alter NMDAR trafficking and synaptic retention, and consequently NMDAR-dependent synaptic plasticity and cognitive tasks[32, 35–37].

We hypothesize that NMDAR-Ab-related psychosis results from a specific alteration of NMDAR trafficking rather than an intrinsic change of the channel activity. Here we investigate the presence and impact of NMDAR-Ab purified from a cohort of healthy subjects and SCZ patients, excluding cases of autoimmune encephalitis. We use classical and high-resolution imaging approaches was implemented in rat hippocampal neurons to provide in-depth information, at the single molecule level, of the impact of NMDAR-Ab on the surface dynamics and nanoscale organization of NMDAR. We show that circulating NMDAR-Ab are detected in about 20% of psychotic patients diagnosed with SCZ and only in very few healthy subjects. Furthermore, NMDAR-Ab from patients, but not from healthy subjects, specifically alter the surface dynamics and nanoscale organization of NMDAR.

synaptic NMDAR, preventing long-term potentiation (LTP) at glutamatergic synapses.

## Results

**Demographic and clinical features of seropositive patients.** Forty-eight SCZ patients were recruited during their hospitalization and included after approval by a French ethical committee and written informed consent for their participation (Supplementary Table 1). A healthy control group ($n = 104$) with no personal or family history of SCZ or bipolar disorder was matched with the SCZ patient sample for age, gender, and years of education (Supplementary Table 1). For all groups, history of stroke, multiple sclerosis, epilepsy, or encephalitis constituted exclusion criteria. The mean age of onset of SCZ was 24.5 years and the mean duration of illness was 11 years (Supplementary Table 2). Most of SCZ patients were under an antipsychotic treatment (74.5%) and eight also received a mood stabilizer (Supplementary Tables 1 and 3). The mean total score for the Positive and Negative Syndrome Scale (PANSS) was 68.6. After performing a series of serological tests to detect the presence of circulating NMDAR-Ab (see section below, Fig. 1a), positive ($n = 9$) and negative ($n = 39$) patients' populations were identified and compared with each other's (Fig. 1a and Supplementary Table 1). Both groups were similar for age, gender, body mass index, age of onset, duration of illness, and number of psychotic episodes. Both seronegative and seropositive patients were under atypical antipsychotic medication, 89% for seronegative and 100% for seropositive patients. Noteworthy, 78% of seropositive patients and only 61.5% of seronegative patients were in acute state (Supplementary Table 2). Seropositive patients' symptoms were consistently more severe than seronegative patients as shown by the significant difference in PANSS total score, PANSS general psychopathology, and PANSS positive (Supplementary Table 2). The clinical files of the nine seropositive patients were systematically reviewed to search for signs of past autoimmune encephalitis (e.g., abnormal movements, bad tolerance to antipsychotics, epilepsy, dysautonomia, and brain magnetic resonance imaging (MRI)). None of these signs or clinical history of encephalitis was found. The infectious screening and detection of other antibodies in seronegative and seropositive patients revealed no obvious alteration, although seropositive patients have other autoantibodies (Supplementary Table 1). Furthermore, as autoantibody titers can fluctuate over time in patients[10, 11], five of the nine seropositive patients were re-hospitalized to perform further medical explorations at a second time point (Supplementary Table 3), while they were outpatient, treated, compliant but remained symptomatic for psychosis. All were re-confirmed for NMDAR-Ab seropositivity. In addition, no neurological symptom was observed, electroencephalogram were unspecific and brain MRI normal. None of them had history of memory disturbance, epilepsy, dysautonomia, abnormal movement, or catatonia. Finally, all the CSF from these patients were negative for NMDAR-Ab (Supplementary Table 3). Altogether, the clinical examination of the seropositive patients fell into a diagnosis of SCZ and failed to detect clinical signs of encephalitis. The seropositive SCZ patients were herein referred as PSY +.

**Detection of circulating NMDAR-Ab.** The detection of circulating NMDAR-Ab in neuropsychiatric patients has been vividly debated over the past years[25]. As mentioned above, we investigated the presence of NMDAR-Ab in the serum of patients and healthy subjects by using a unique combination of conventional and single-molecule high-resolution imaging approaches. First, we used classical cell-based assays on live HEK cells ectopically expressing GluN1-GFP and GluN2 NMDAR subunits (Fig. 1a).

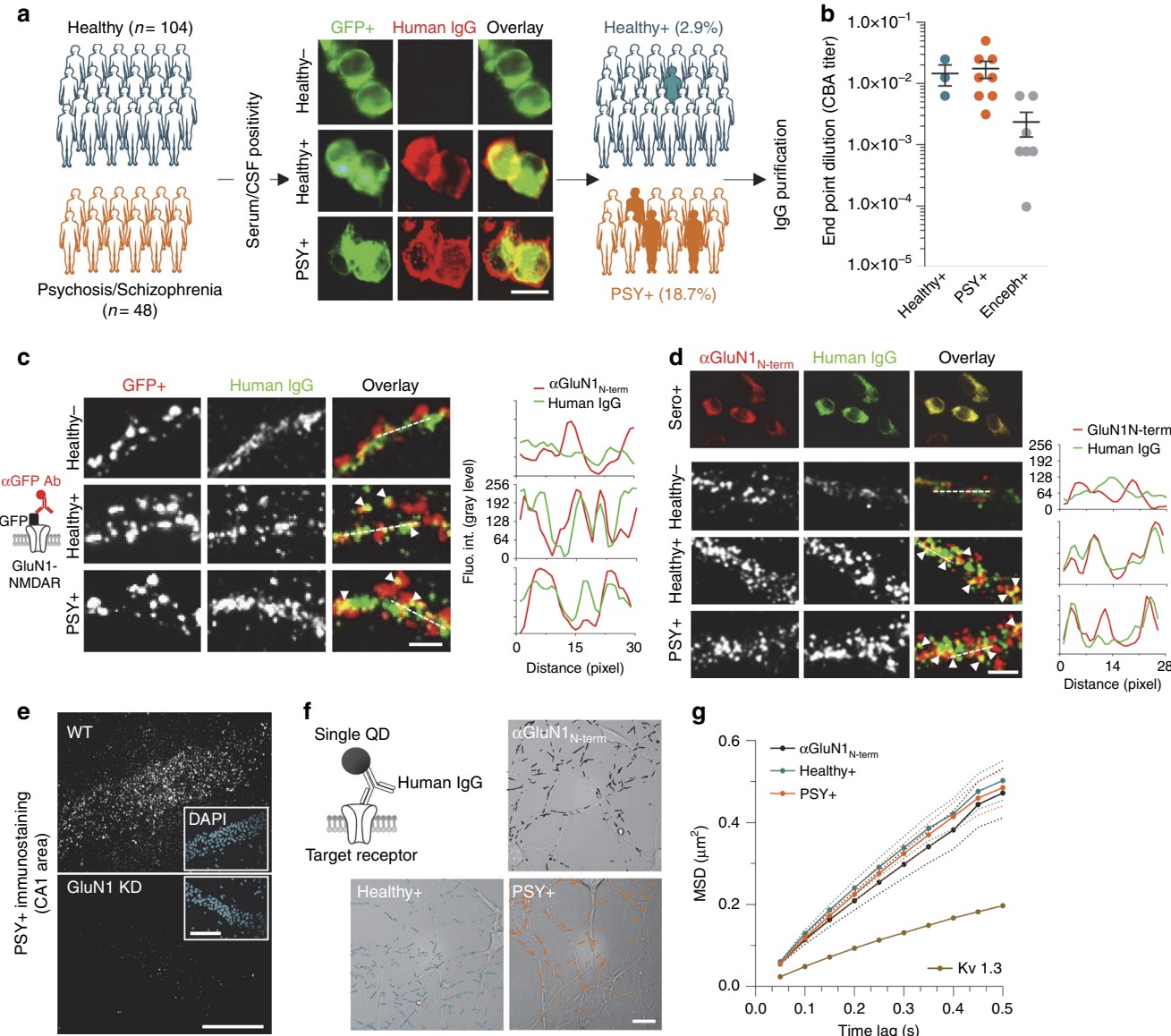

**Fig. 1** Multi-approach identification of NMDAR-Ab IgG in the serum of schizophrenia patients and healthy subjects. **a** Immunostaining of HEK293 cells expressing GluN1-GFP and GluN2B subunits labeled with the sera of healthy subjects or schizophrenic patients (1/10, 3 h incubation). Note the overlap between serum reactivity (red) and GFP-positive HEK cells (green) for seropositive samples. Scale bar, 10 μm. **b** Mean ± SEM of NMDAR-Ab titers estimated by end-point dilutions for Healthy + (n = 3 samples), PSY + (n = 8) and Enceph + (n = 7) serum samples. Each data point represents the value calculated for a subject or a patient. **c** Surface co-immunostaining for GluN1-SEP containing NMDAR (red) and human purified IgG's target (green) in live hippocampal neurons. Both Healthy + and PSY + IgG detect a target that colocalizes with surface GluN1-SEP clusters (white arrowheads). Scale bar, 2 μm. Right panel: corresponding linescans plotting GFP + and human IgG fluorescence intensities over distance (pixel). **d** Upper panel: surface co-immunostaining of HEK293 cells with a commercial anti-GluN1 antibody targeting the extracellular part of the GluN1 subunit and seropositive samples. Lower panel: representative dendritic areas of cultured hippocampal neurons (12 div) co-labeled with purified IgG (5 μg ml⁻¹, green) from Healthy −, Healthy +, or PSY + subjects and a commercial anti-GluN1 antibody targeting the extracellular part of the GluN1 subunit (αGluN1$_{N-term}$, red). Scale bar, 2 μm. Right panel: corresponding linescans plotting endogenous GluN1-NMDAR and human IgG fluorescence intensities over distance (pixel). **e** Immunostaining of CA1 hippocampal sections (20 μm thick) with PSY + IgG from wild-type (WT) and GluN1 knock-down (GluN1-KD) mice. Insets, DAPI staining (blue). Scale bar, 200 μm; scale bar inset, 100 μm. **f** Representative trajectories obtained with αGluN1$_{N-term}$ (black lines), Healthy + (blue) and PSY + (orange) IgG-QD complexes used to label the surface target of patients' IgG. IgG-QD complexes were tracked during 500 frames with a 50ms acquisition frequency on cultured hippocampal neurons (14–15 div). Scale bar, 20 μm. **g** Mean Square Displacement (MSD) over time of endogenous GluN1-NMDAR targeted by a commercial αGluN1$_{N-term}$ antibody (n = 5 neurons) or using purified IgG from Healthy + (n = 6) or PSY + (n = 10) individuals. The curves are represented as mean ± SEM (dash lines)

These assays were independently duplicated in two laboratories (Lyon, France and Barcelona, Spain). Sera from nine PSY + patients and three healthy individuals (Healthy + ) stained green fluorescent protein (GFP)-positive HEK cells contrary to seronegative samples (Healthy − ) (Fig. 1a and Supplementary Table 1). Autoantibody titers were then estimated as end-point dilutions, ranging between 1/20 and 1/320 with no significant difference between PSY + patients and Healthy + subjects (Fig. 1b). When these values were compared to the ones obtained from gold-standard anti-NMDAR encephalitis patients, a

significant 20-fold decrease was observed (mean end point dilution: 1/3200; Fig. 1b). It could be noted that end point dilutions of few PSY + and encephalitis patients overlap, suggesting that

although levels of circulating NMDAR-Ab in PSY + patients and Healthy + subjects are significantly lower than in anti-NMDAR encephalitis patients, a fraction of patients with distinct clinical

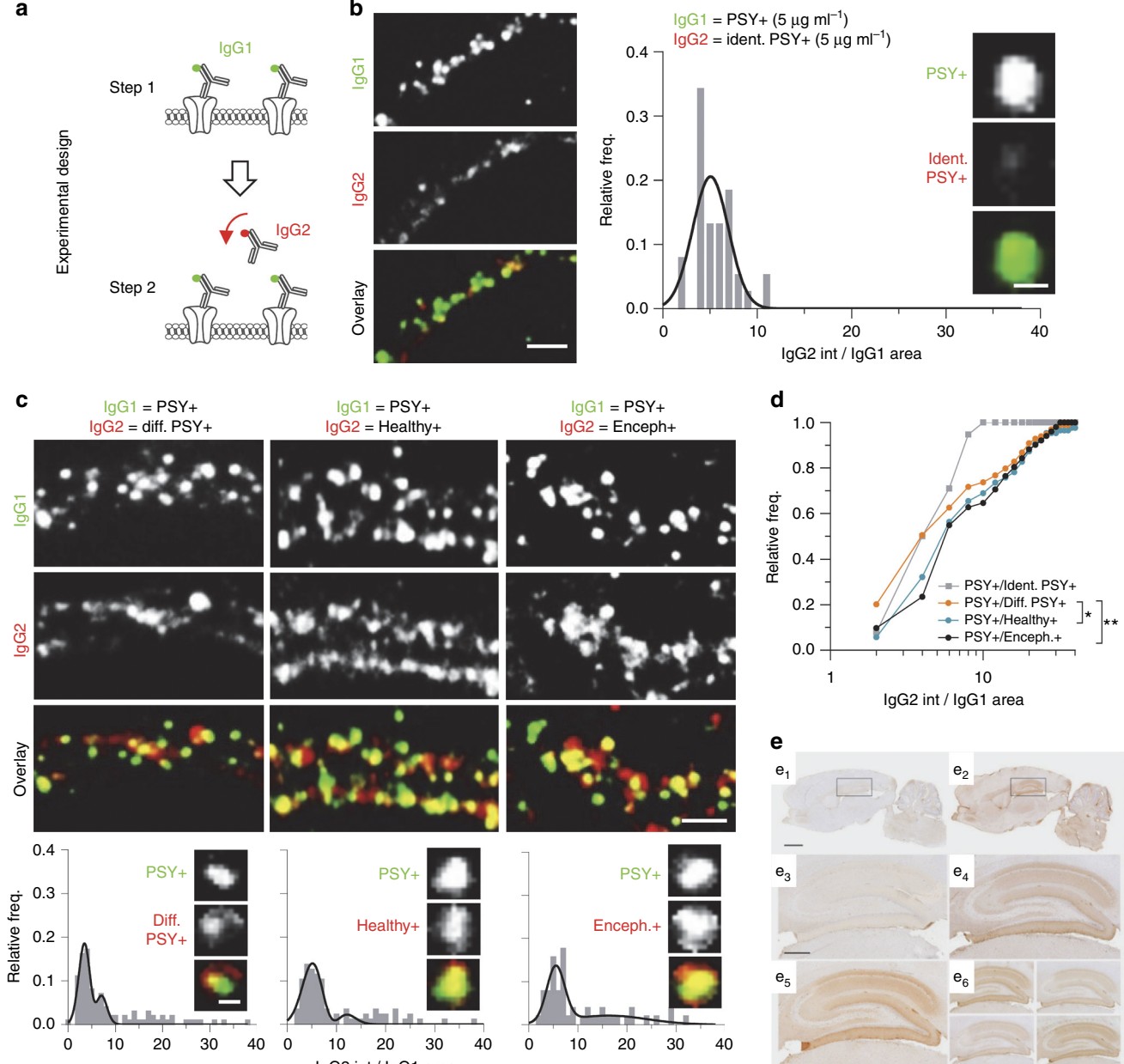

**Fig. 2** NMDAR-Ab from different origins target do not compete for target binding. **a** Experimental design of the in vitro immuno-competition test. Hippocampal cultures (12 div) were first incubated with PSY + NMDAR-Ab (IgG1) labeled in green. Cells were then incubated with a second Healthy +, PSY + or Enceph + IgG (IgG2) labeled in red. Three out of three Healthy +, four out of nine PSY +, and one out of seven encephalitis purified IgG samples were used and pooled for comparisons. **b** Representative dendritic areas labeled with PSY + NMDAR-Ab (IgG1, 5 µg ml$^{-1}$, green) and treated with the same IgG (IgG2 = ident. PSY +, 5 µg ml$^{-1}$, red). Scale bar, 2 µm. Right panel: histogram of IgG2 fluorescence intensity within IgG1 area. The use of the same IgG (IgG2 = ident. PSY +) results in a distribution fitted with a single Gaussian. Insets, staining from a single cluster. Scale bar, 500 nm. **c** Representative dendritic areas labeled with a PSY + NMDAR-Ab (IgG1, 5 µg ml$^{-1}$, green) and treated with NMDAR-Ab from another PSY + patient (IgG2 = Diff. PSY +), a healthy individual (IgG2 = Healthy +) or a patient with anti-NMDAR encephalitis (IgG2 = Enceph +). Scale bar, 2 µm. Bottom panels: Corresponding histograms of IgG2 fluorescence intensity within IgG1 area for the different competing conditions. Insets, staining from single clusters. Scale bar, 500 nm. **d** Cumulative distributions of IgG2 intensity within IgG1 cluster areas according to the different competing conditions: PSY + /Ident. PSY + (n = 38 dendritic regions, N = 10 neurons), PSY + /Diff. PSY + (n = 99, N = 32), PSY + /Healthy + (n = 78, N = 26) or PSY + /Enceph + (n = 51, N = 15); *P < 0.05, **P < 0.005, Kolmogorov–Smirnov test. **e** Competing biotinylation assay in rat hippocampal slices. Pre-incubation of rat brain sections with serum from a patient with anti-NMDAR encephalitis blocks the reactivity of biotinylated IgG from another patient with anti-NMDAR encephalitis (left and middle panels). Pre-incubated with serum from a healthy control does not block the reactivity of biotinylated IgG from a patient with anti-NMDAR encephalitis (e2 and e4). Pre-incubation of sections of rat brain with serum of five patients with schizophrenia does not block the reactivity of biotinylated IgG from a patient with anti-NMDAR encephalitis (e5 and e6)

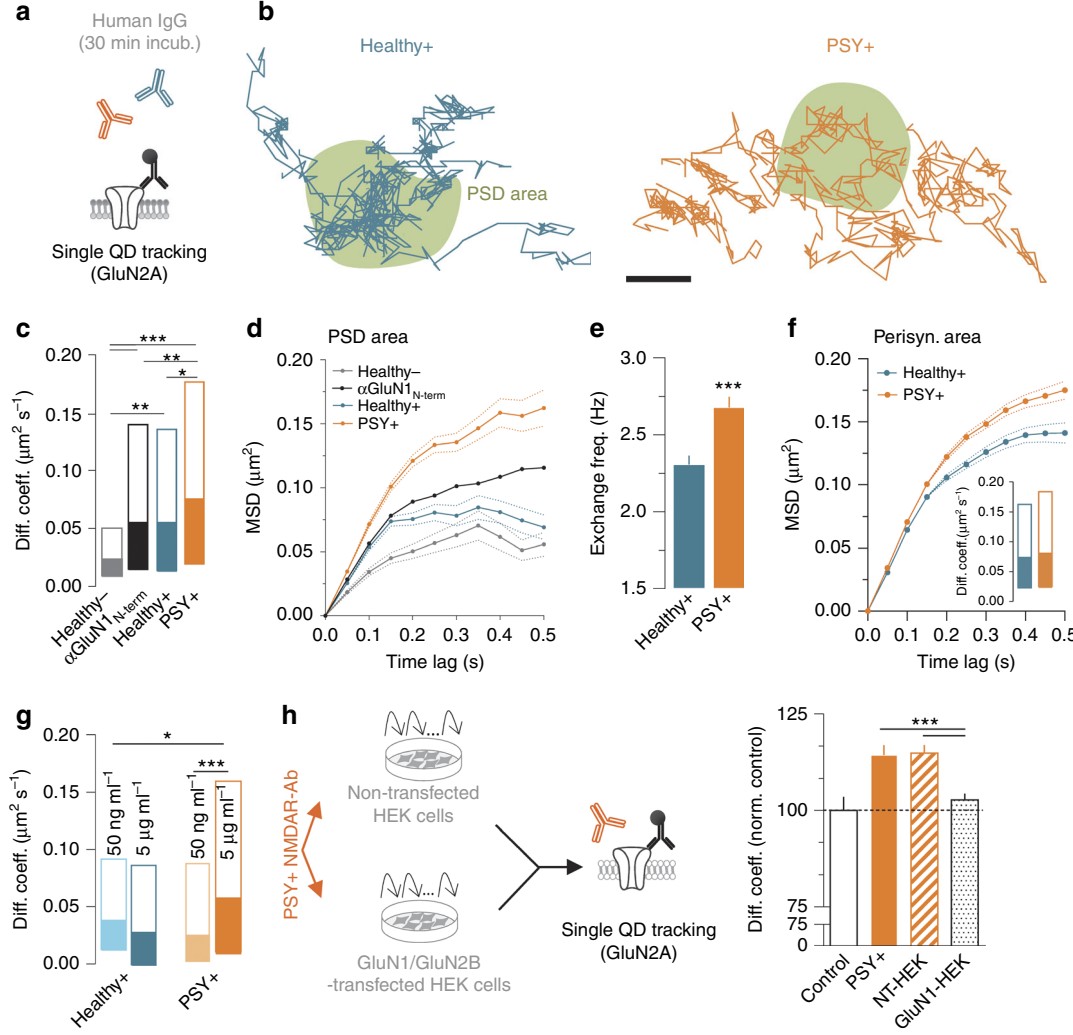

**Fig. 3** NMDAR-Ab from Healthy + subjects and PSY + patients display different effects on synaptic NMDAR dynamics. **a** Schematic representation of the experimental design. **b** Representative trajectories of a single GluN2A-NMDAR-QD complex (500 frames, 50 ms acquisition) exposed for 30 min to purified IgG (5 μg ml$^{-1}$) from Healthy + or PSY + subjects within synaptic areas. Scale bar, 500 nm. **c** Comparison of GluN2A-NMDAR instantaneous diffusion coefficient (μm$^2$ s$^{-1}$) within the postsynaptic density (PSD) area in the presence of a commercial αGluN1$_{N-term}$ antibody (n = 749 trajectories, N = 52 neurons) or purified IgG from Healthy − (n = 94, N = 13), Healthy + (n = 313, N = 42) or PSY + (n = 695, N = 59) subjects (***P < 0.0001, Kruskal–Wallis followed by Dunn's multiple comparison test). Data are expressed as median diffusion coefficient ± 25–75% IQR. **d** Mean square displacement (MSD) over time of GluN2A-NMDAR within the PSD area after exposure to a commercial αGluN1$_{N-term}$ antibody or purified IgG from Healthy −, Healthy +, and PSY + individuals (P < 0.001, Kolmogorov–Smirnov test). Data are expressed as mean ± SEM (dash lines). **e** Mean ± SEM of diffusive GluN2A-NMDAR exchange frequency between synaptic and extrasynaptic compartments in Healthy + (n = 241 trajectories) or PSY + (n = 268) conditions (***P < 0.001, Mann–Whitney test). **f** MSD over time of GluN2A-NMDAR within the perisynaptic area exposure to Healthy + or PSY + NMDAR-Ab for 30 min (P = 0.076, Kolmogorov–Smirnov test). Data are expressed as mean ± SEM (dash lines). Inset: GluN2A-NMDAR instantaneous diffusion coefficient within the perisynaptic area in Healthy + (n = 1344 trajectories, N = 42 neurons) or PSY + (n = 2264, N = 59) conditions (P = 0.213, Mann–Whitney test). **g** Comparison of GluN2A-NMDAR instantaneous diffusion coefficient after 30 min exposition to various concentrations of NMDAR-Ab. Data are expressed as median diffusion coefficient ± 25-75% IQR. Healthy + 50 ng ml$^{-1}$, n = 321 trajectories, N = 21 neurons; Healthy + 5 μg ml$^{-1}$, n = 69, N = 32; PSY + 50 ng ml$^{-1}$, n = 187, N = 24; PSY + 5 μg ml$^{-1}$, n = 178, N = 32. *P < 0.05, ***P < 0.001, Kruskal–Wallis followed by Dunn's multiple comparison test. **h** Experimental design of the NMDAR-Ab pre-absorption experiment. Right panel: normalized GluN2A instantaneous diffusion coefficient in control (n = 1,388 trajectories, N = 37 neurons), PSY + (n = 2,658, N = 59), NT-HEK (n = 3867, N = 27), and GluN1-HEK (n = 4799, N = 28) conditions (***P < 0.001, Kruskal–Wallis followed by Dunn's multiple comparison test)

features have similar level of circulating NMDAR-Ab. Next, we tested the sample seropositivity by using a variant of the cell-based assay, i.e., fixed HEK cells overexpressing GluN1-NMDAR, which is a sensitive and reliable method used to detect NMDAR-Ab in anti-NMDAR encephalitis patients[26, 38]. None of the samples were positive using this method. Thus, detection methods that are reliable to detect NMDAR-Ab in anti-NMDAR encephalitis patients provide contradictory outcomes when used

to detect NMDAR-Ab from PSY + patients or Healthy + subjects, strengthening the urgent need for new detection methods[27].

To further confirm the presence of NMDAR-Ab, we performed a live double immunostaining of GluN1-SEP, a genetically modified GluN1 subunit containing a supereclipic GFP (SEP) at its extracellular N-terminus, and purified IgG from seronegative and seropositive samples in dissociated hippocampal neurons (Fig. 1c). Surface clusters of GluN1-SEP colocalized with Healthy + and PSY + IgG staining but not with Healthy −, as exemplified

in the corresponding fluorescence intensity curves (Fig. 1c). Consistently, the live staining obtained with Healthy + and PSY + IgG predominantly overlapped with the distribution of endogenous membrane NMDAR (Fig. 1d), further supporting the presence of circulating NMDAR-Ab in PSY + patients and Healthy + subjects. A comparative labeling was also observed between endogenous NMDAR and NMDAR-Ab in hippocampal slices, strengthening previous in vitro observations (Supplementary Fig. 1d). Additional labeling revealed that Healthy + and PSY + purified IgG do not target other proteins present at glutamatergic synapses such as IgLON5 (Supplementary Fig. 1a) or GluA1-AMPAR (Supplementary Fig. 1b) although they are found in synaptic areas (Supplementary Fig. 1c). Finally, PSY + purified IgG were used to immunostain hippocampal brain sections from either wild-type or GluN1 knockdown mice[39] (GluN1-KD, expressing only 10% of GluN1 subunit). As expected from IgG directed against the GluN1 subunit, there was virtually no immunostaining in the CA1 area from the GluN1 KD tissue when compared with the wild-type one (Fig. 1e). Altogether, these immunostaining assays support the presence of NMDAR-Ab in both healthy subjects and schizophrenic patients.

In search for an additional sensitive assay in living neurons, we took advantage of the single-molecule imaging, as the molecular behavior of an individual membrane receptor (targeted by an antibody) is given by robust biophysical characteristics and is virtually independent of the antibody concentration[40–42]. The target of the autoantibodies was examined by coupling together purified IgG from Healthy + or PSY + individuals to single nanoparticles (Quantum dots, QD) (Fig. 1f). This autoantibody–target complex was imaged with a sub-wavelength precision at video rate in live hippocampal cultured networks (Fig. 1f), giving access to the live "signature" of the Healthy + subjects' and PSY + patients' IgG target[40, 41]. The mean square displacement (MSD) curves, reflecting the surface explored and the type of motion of the complex, revealed that the signatures of endogenous GluN1-NMDAR, Healthy + subjects and PSY + patients were undistinguishable whereas the potassium channel Kv1.3 MSD was clearly different (Fig. 1g). Collectively, our combination of conventional macroscopic and single nanoparticle imaging approaches provides robust evidence that NMDAR-Ab are present in the serum of a subset of schizophrenic patients and very few healthy subjects.

**PSY + NMDAR-Ab do not compete for target binding**. To investigate whether NMDAR-Ab from Healthy + subjects and PSY + patients similarly bind NMDAR, we performed immuno-competition assays in dissociated hippocampal neurons and acute slices. For the in vitro assay, neurons were first incubated with NMDAR-Ab from PSY + patients followed either by identical PSY + NMDAR-Ab (i.e., from the same patient), different PSY + NMDAR-Ab, Healthy + NMDAR-Ab, or encephalitis NMDAR-Ab (Figs. 2a, b). From now on and as detailed in figure legends, purified IgG from three out of three Healthy + subjects and four out of nine PSY + patients were used to perform in vitro experiments (samples from five PSY + patients were not used, as the available material was not sufficient to purify IgGs). Based on a dose-dependent competition assay and fluorescence background measurements (Supplementary Fig. 2), we selected 5 μg ml$^{-1}$ IgG concentration for the immuno-competition assay. The fluorescence intensity of the secondary staining (i.e., IgG2) was measured within the area of the primary staining (i.e., IgG1). As expected, identical IgG1 and IgG2 highly compete, leading to a very low IgG2 staining and unimodal distribution of the IgG2/IgG1 ratio (Fig. 2b). Strikingly, little, if any, competition was observed between different PSY + NMDAR-Ab, Healthy +

NMDAR-Ab, and encephalitis NMDAR-Ab (Fig. 2c). A significant shift in the cumulative distributions of IgG2/IgG1 ratio was observed between PSY + patients and Healthy + NMDAR-Ab, as well as between PSY + and encephalitis patients (Fig. 2d), indicating that NMDAR-Ab of various origins differently bind NMDAR. Next, we confirmed part of these data in brain tissue by performing a competition assay in which acute hippocampal slices were first incubated with either PSY + or encephalitis NMDAR-Ab followed by biotinylated encephalitis NMDAR-Ab (Fig. 2e). Encephalitis NMDAR-Ab competed with each other for the same binding site but they failed to compete with PSY + NMDAR-Ab (Fig. 2e). Together, these data provide evidence that NMDAR-Ab from PSY + patients, Healthy + subjects, and encephalitis patients do not compete for binding on their target.

**PSY + NMDAR-Ab alter NMDAR surface dynamics**. Based on the above differences in binding properties, we explored the potency of NMDAR-Ab from Healthy + subjects and PSY + patients to acutely alter NMDAR surface trafficking (Fig. 3a). Indeed, antibodies directed against extracellular epitopes of the NMDAR can acutely alter the surface distribution and dynamics of the receptors[32, 36, 43]. We specifically investigated the impact of autoantibodies on synaptic NMDAR by focusing on the GluN2A-NMDAR subtype located within postsynaptic density areas (PSD areas). Patients' IgG (one Healthy –, three Healthy +, and four PSY +) were tested separately. Noteworthy, only 6% variability was found between the different PSY + patients (Supplementary Fig. 3). Strikingly, NMDAR-Ab from PSY + patients increased GluN2A-NMDAR surface dynamics (Fig. 3b) when compared with Healthy + subjects or to a commercial anti-GluN1 antibody (Fig. 3c). Consistently, PSY + NMDAR-Ab shifted up GluN2A-NMDAR MSD curves, indicating a lower confinement and a larger explored area (Fig. 3d). In addition, the exchange frequency, i.e., the number of receptor entries and exits between the synaptic area and its periphery, was significantly increased by PSY + NMDAR-Ab (Fig. 3e). This robust effect was mainly observed within the synaptic area as only a tendency was observed within the perisynaptic area (Fig. 3f). Noteworthy, the presence of an anti-GluN1 antibody or Healthy + IgG slightly increased the receptor surface trafficking, consistent with the steric hindrance induced by an antibody within the synaptic cleft[41]. Collectively, these data provide direct evidence, at the single-molecule level that NMDAR-Ab from different origins produce distinct molecular effects on the synaptic trafficking of surface NMDAR.

To assess the direct role of NMDAR-Ab among the purified IgG, we first tested the impact of a diluted solution of PSY + NMDAR-Ab (50 ng ml$^{-1}$ of purified IgG) on GluN2A-NMDAR dynamics within synaptic areas. Such a 100-fold dilution fully abolished the effect on surface dynamics (Fig. 3g). Then, we specifically reduced the content of PSY + NMDAR-Ab by performing a pre-absorption experiment in which PSY + IgG solutions were repeatedly exposed to either untransfected or transfected (GluN1/GluN2 subunits) live HEK cells (Fig. 3h), as previously published[44, 45]. The IgG solutions collected from untransfected HEK cells increased GluN2A-NMDAR surface dynamics within the synaptic area, whereas no effect was observed in the presence of the IgG solutions collected from GluN1-transfected HEK cells (Fig. 3h). These data confirm that alterations of synaptic GluN2A-NMDAR surface dynamics indeed result from the presence of NMDAR-Ab from PSY + patients. Finally, we tested whether PSY + NMDAR-Ab effect was specific for NMDAR by investigating their impact on another receptor, i.e., GluA1-AMPA receptor (GluA1-AMPAR), and a potassium channel, i.e., Kv1.3 type, which are both present at the

glutamatergic synapse and interact with similar scaffolds in the synapse[46]. PSY + NMDAR-Ab neither altered the diffusion coefficients nor the MSD curves of GluA1-AMPAR and Kv1.3

channel (Supplementary Fig. 4). Altogether, these data indicate that NMDAR-Ab from PSY + patients, and not Healthy + subjects, specifically and rapidly "destabilize" synaptic NMDAR.

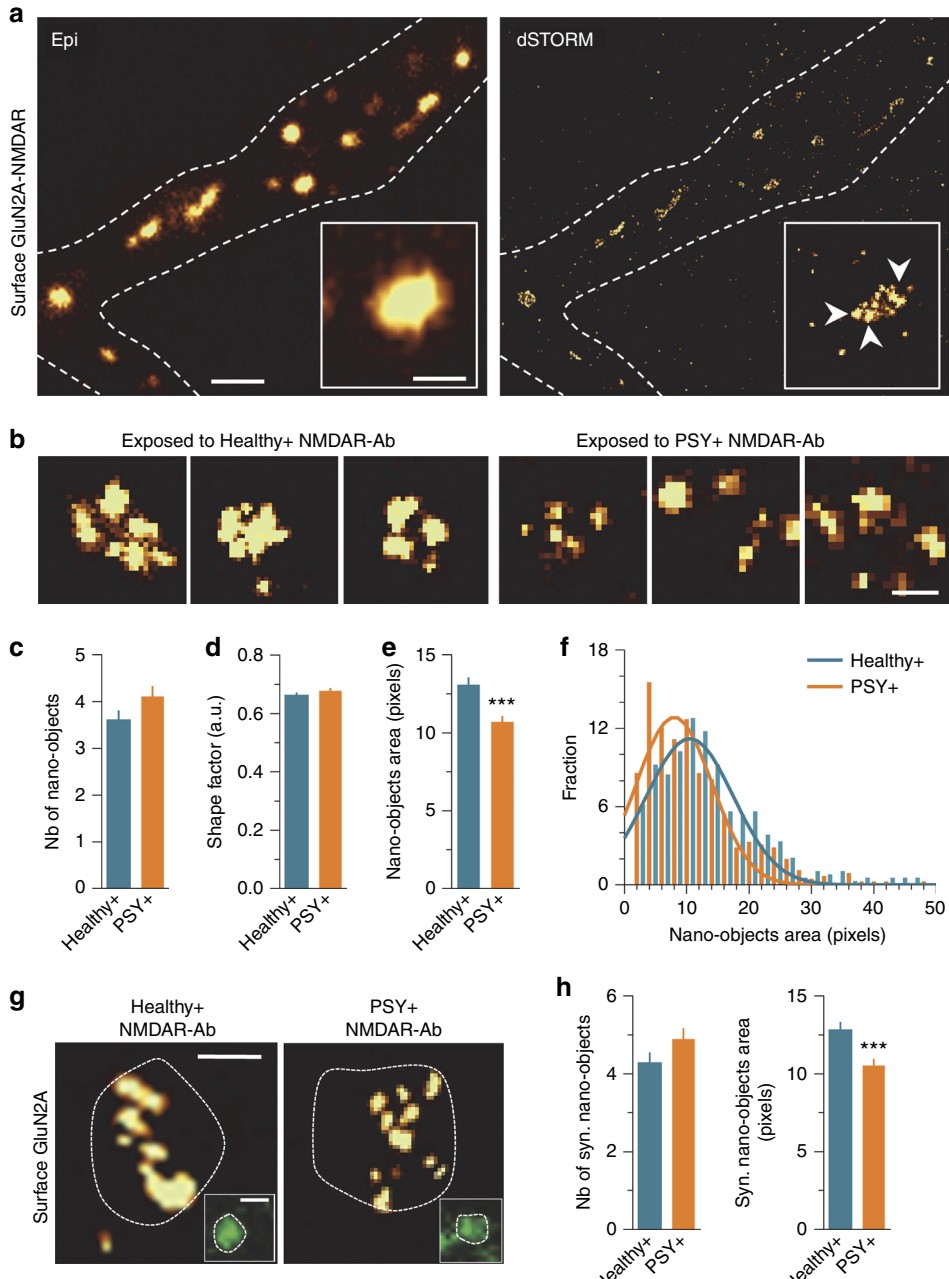

**Fig. 4** Super-resolved map of GluN2A-NMDAR surface organization in presence of Healthy + and PSY + NMDAR-Ab. **a** Epifluorescence and dSTORM image of a dendritic segment with endogenous GluN2A-NMDAR staining. Scale bar, 1 μm. Insert panel, magnification of an isolated GluN2A-NMDAR cluster. Scale bar, 300 nm. **b** Representative examples of GluN2A-NMDAR nano-objects imaged by dSTORM after incubation with either Healthy + or PSY + NMDAR-Ab. One out of three Healthy + and 1/9 PSY + purified IgG samples were used and pooled for comparisons. Scale bar, 400 nm. **c** Mean ± SEM of GluN2A-NMDAR nano-object numbers after exposure to Healthy + ($n = 109$ nano-objects, $N = 4$ neuronal fields) or PSY + ($n = 112$, $N = 5$) NMDAR-Ab ($P = 0.0751$, Mann–Whitney test). **d** Mean ± SEM of GluN2A-NMDAR nano-object shape factor in Healthy + ($n = 368$ nano-objects) and PSY + ($n = 417$) conditions ($P = 0.3374$, Mann–Whitney test). **e** Mean ± SEM of GluN2A nano-object area in Healthy + ($n = 392$ clusters) or PSY + ($n = 458$) conditions. Note that PSY + decrease GluN2A nano-objects area compared with Healthy + NMDAR-Ab (***$P < 0.001$, Mann–Whitney test). **f** Distribution of GluN2A nano-objects area in Healthy + or PSY + conditions. Note the homogenous left shift of GluN2A nano-objects area in the presence of PSY + NMDAR-Ab. **g** Epifluorescence image of PSD-95 staining and dSTORM image of GluN2A subunits in neurons exposed to Healthy + or PSY + NMDAR-Ab. PSD-95 staining was used to delineate the synaptic area (white dashed lines). PSD-95 scale bar, 400 nm. GluN2A scale bar, 200 nm. **h** Mean ± SEM of synaptic GluN2A-NMDAR nano-objects number (Healthy + , $n = 70$ nano-objects; PSY + , $n = 66$; $P = 0.0937$, Mann–Whitney test) and area (Healthy + , $n = 301$ nano-objects; $PSY + $, $n = 322$) between Healthy + and PSY + NMDAR-Ab conditions. Note the reduced synaptic GluN2A nano-objects area in the presence of PSY + compared with Healthy + NMDAR-Ab (***$P < 0.001$, Mann–Whitney test)

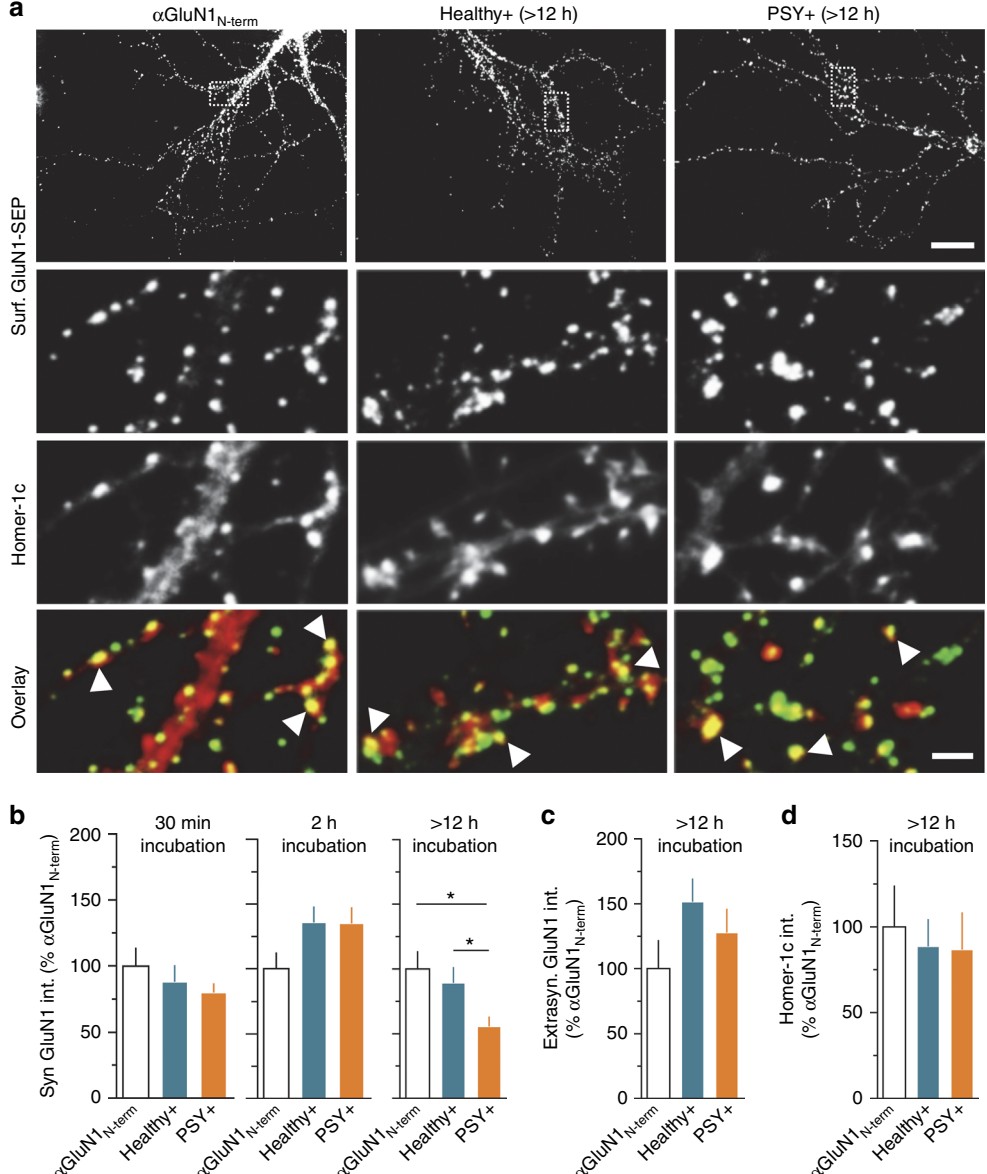

**Fig. 5** PSY + NMDAR-Ab specifically alter NMDAR synaptic content in a time-dependent manner. **a** Live staining of GluN1-SEP in hippocampal cultures treated overnight with either a control antibody (αGluN1$_{N-term}$) or human purified IgG (Healthy + or PSY + NMDAR-Ab). The synaptic localization of NMDAR clusters (green) was determined using Homer-1c (red). Colocalized staining was interpreted as synaptic GluN1-NMDAR clusters (white arrowheads). Three out of three Healthy + and four out of nine PSY + samples were used and pooled for comparisons. Scale bar, 20 μm (upper panels), 2 μm (lower panels). **b** Normalized synaptic GluN1-NMDAR fluorescence intensity (normalization to the control condition αGluN1$_{N-term}$) after incubation with αGluN1$_{N-term}$ ($n = 19$ neurons), Healthy + ($n = 36$), or PSY + ($n = 55$) for 30 min ($P = 0.7604$, one-way ANOVA); after incubation with αGluN1$_{N-term}$ ($n = 24$), Healthy + ($n = 47$) or PSY + ($n = 77$) for 2 h ($P = 1.389$, one-way ANOVA), or after incubation with αGluN1$_{N-term}$ ($n = 22$), Healthy + ($n = 34$) or PSY + ($n = 28$) for 12 h. Synaptic NMDAR fluorescence intensity is reduced after a 12 h exposure to PSY + NMDAR-Ab (*$P < 0.05$, one-way ANOVA followed by Tukey's multiple comparison test). **c** Normalized extrasynaptic GluN1-NMDAR fluorescence intensity after 12 h incubation with αGluN1$_{N-term}$ ($n = 14$ neurons), Healthy + ($n = 34$), or PSY + ($n = 30$) NMDAR-Ab ($P = 0.2385$, Kruskal–Wallis test). **d** Normalized Homer-1c fluorescence intensity after a 12 h incubation with αGluN1$_{N-term}$ ($n = 25$ neurons), Healthy + ($n = 31$), or PSY + ($n = 16$) NMDAR-Ab ($P = 0.1073$, one-way ANOVA)

**PSY + NMDAR-Ab alter NMDAR nanoscale organization**. An altered surface dynamics of a given receptor may, or may not, lead to changes of its synaptic organization and content[42]. To investigate the fine organization of NMDAR within synapses, we used super-resolution microscopy, as it overcomes the diffraction limit and constitutes a powerful approach to reveal the molecular organization of postsynaptic molecules at nanoscale resolution[47]. To test whether NMDAR-Ab alter the organization of NMDAR, in a uniform or non-uniform manner, we took advantage of the direct stochastic optical reconstruction microscopy (dSTORM)

imaging to obtain the first super-resolved map of surface endogenous GluN2A-NMDAR (Fig. 4a). Initially, endogenous GluN2A-NMDAR were organized in three to four juxtaposed nano-objects (Fig. 4b), resembling the organization of other glutamate receptors[48, 49]. In neurons exposed to PSY + NMDAR-Ab, the mean number and shape of NMDAR nano-objects remained unaffected, whereas the nano-objects area was reduced (Figs. 4c–e). Indeed, the nano-objects size distribution was homogeneously shifted by PSY + NMDAR-Ab, suggesting that all nano-objects were similarly affected (Fig. 4f). In order to

specifically characterize NMDAR nanoscale organization within glutamatergic synapses, PSD-95, a core protein of the PSD, was co-immunolabeled (Fig. 4g). The nano-objects area was consistently reduced in PSD-95-containing synapses exposed to PSY + NMDAR-Ab without any effect on their number (Fig. 4h). Together, these data reveal that PSY + NMDAR-Ab reduce the area of synaptic NMDAR nano-objects, while leaving unaltered the nano-object organization (i.e., number and shape), supporting a model in which PSY + NMDAR-Ab uniformly decrease the content of NMDAR within synaptic nano-objects.

Next, we investigated whether this synaptic nanoscale reorganization translated to a decrease of NMDAR synaptic content. Using immunocytochemical staining and confocal microscopy, we quantified the surface NMDAR content in synapses by immunostaining GluN1-SEP in live neurons after a 30 min- (as the acute effect on surface dynamics), 2 h-, or 12 h-incubation period with the different NMDAR-Ab (Fig. 5). At all incubation times, hippocampal neurons exposed to Healthy + NMDAR-Ab exhibited similar intensities of synaptic NMDAR cluster when compared with a control condition (incubation with $\alpha$GluN1$_{\text{N-term}}$) (Fig. 5b). Of note, neither synaptic nor extra-synaptic NMDAR content were affected after 12 h exposure to purified IgG from Healthy−subjects (Supplementary Fig. 5c). In contrast, PSY + NMDAR-Ab significantly reduced GluN1-

NMDAR synaptic content following a 12 h incubation period, whereas shorter incubation time had no effect (Figs. 5a, b). The decrease after 12 h incubation was restricted to the synaptic pool, as GluN1-NMDAR extrasynaptic content remained stable (Fig. 5c and Supplementary Fig. 5a, b). The PSD of glutamatergic synapses, estimated by the detection of Homer-1c, remained unaltered in all conditions (Fig. 5d and Supplementary Fig. 5d). Altogether, these data indicate that PSY + NMDAR-Ab specifically reduce GluN1-NMDAR synaptic nanoscale organization and content over time, leading to the loss of nearly half the receptors after several hours.

**PSY + NMDAR-Ab disorganize EphrinB2 receptor within synapses.** The destabilization and lateral displacement of synaptic receptors often originate from the disruption of interactions with anchoring partners, such as transmembrane receptors or scaffold proteins[42]. As NMDAR-Ab bind extracellular epitope(s) of the receptor and can thus perturb interactions between NMDAR and an anchoring partner, we primarily focused our attention on the EphrinB2 receptor (EphB2R), as it strongly retains synaptic NMDAR through a direct interaction of their respective extracellular domains[50]. We first performed immunocytochemical detection of surface EphB2R in neurons exposed to purified IgG

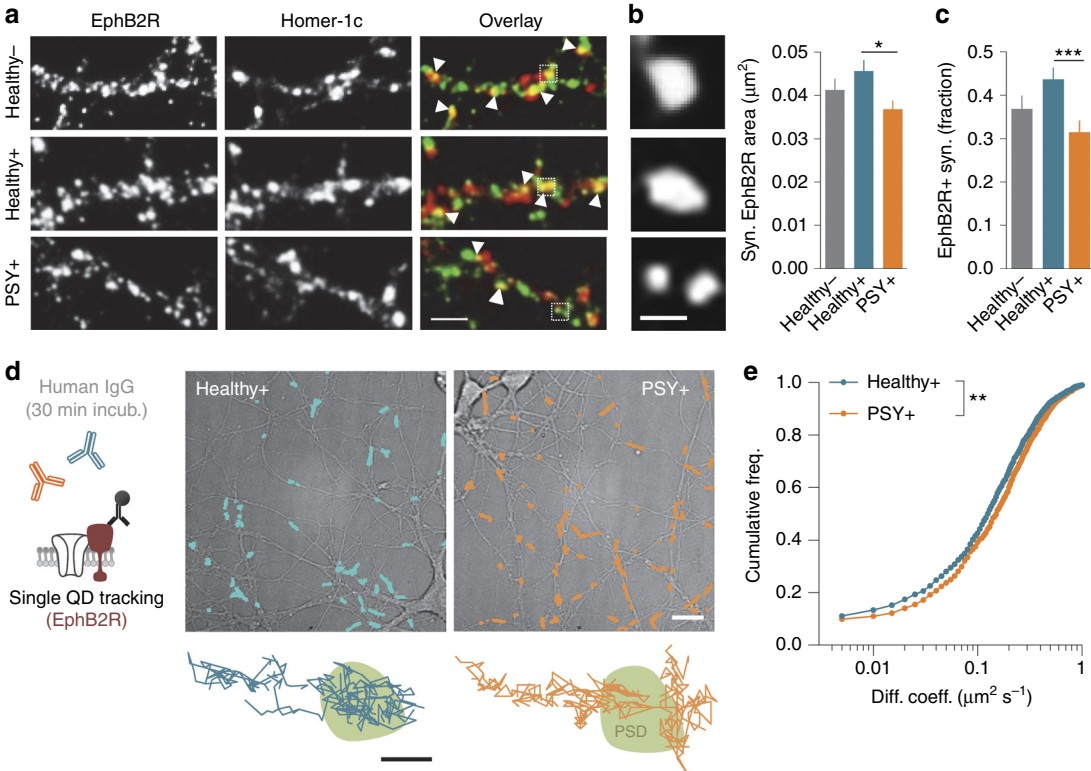

**Fig. 6** Synaptic EphB2R diffusion and distribution are specifically altered by PSY + NMDAR-Ab. **a** Immunostaining of surface EphB2R in hippocampal neurons (12 div) exposed to purified IgG from Healthy −, Healthy +, or PSY + samples for 12 h. One out of one Healthy −, three out of three Healthy +, and four out of nine PSY + samples were used and pooled for comparisons. Synaptic EphB2R were determined as overlapping clusters with the synaptic marker Homer-1c (white arrowheads). Scale bar, 2 μm. **b** Left panel: synaptic EphB2R clusters after a 12 h exposure to Healthy −, Healthy +, or PSY + purified IgG. Scale bar, 500 nm. Right panel: mean ± SEM of synaptic EphB2R clusters area exposed to Healthy − (n = 24 neurons), Healthy + (n = 49), or PSY + (n = 40) purified IgG. Exposure to PSY + NMDAR-Ab for 12 h reduces the area of synaptic EphB2R clusters (*P = 0.0359, Kruskal–Wallis test followed by Dunn's multiple comparison test). **c** Fraction of EphB2R-positive synapses after incubation with Healthy − (n = 25 neurons), Healthy + (n = 49), or PSY + (n = 41) purified IgG. PSY + NMDAR-Ab affect the proportion of synaptic EphB2R (***P < 0.001, Kruskal–Wallis test followed by Dunn's multiple comparison test). **d** Surface live tracking of endogenous EphB2R in hippocampal neurons (15 div) exposed to Healthy + or PSY + NMDAR-Ab for 30 min. Scale bar, 20 μm. Bottom panel: Representative EphB2R single trajectories within PSD areas. Scale bar, 500 nm. **e** Cumulative distributions of the instantaneous diffusion coefficient of synaptic EphB2R in the presence of Healthy + (n = 1,432 trajectories, N = 33 neurons) or PSY + NMDAR-Ab (n = 1104, N = 39). The distribution is shifted toward the right after exposure to PSY + NMDAR-Ab (**P < 0.01, Kolmogorov–Smirnov test)

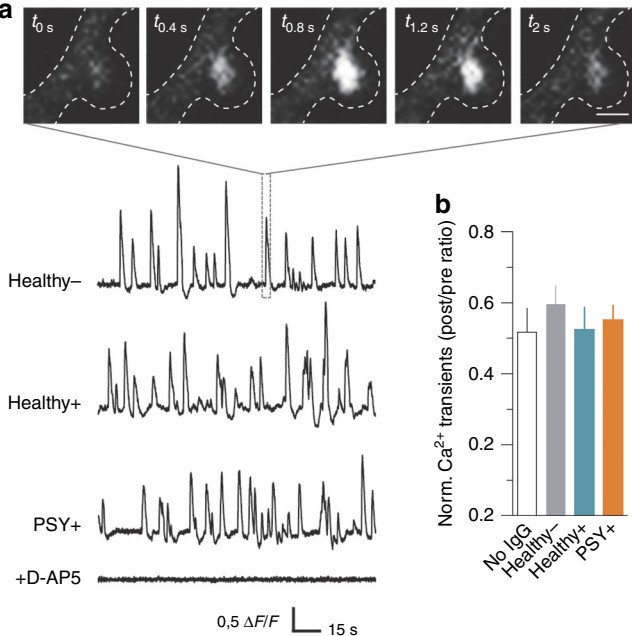

**Fig. 7** NMDAR-Ab from Healthy + subjects and PSY + patients do not affect NMDAR-mediated Ca²⁺ transients in spines of hippocampal neurons. **a** Representative time-lapse images of a spontaneous NMDAR-mediated $Ca^{2+}$ transient in basal condition (in the presence of nifedipine 5 μM and bicuculline 5 μM). Scale bar, 2 μm. Lower panel, representative examples of NMDAR-mediated $Ca^{2+}$ transients recorded in spines (expressed as $\Delta F/F$ ratio) exposed to purified IgG from Healthy −, Healthy +, or PSY + individuals. Note that all events were abolished by the NMDAR-competitive antagonist D-AP5 (50 μM). **b** Normalized frequency (ratio of $Ca^{2+}$ transients frequency post-application of NMDAR-Ab relative to the baseline acquisition) of spontaneous NMDAR-mediated $Ca^{2+}$ transients in control condition with no IgG ($n = 38$ spines, $N = 4$ neurons) or in the presence of Healthy − purified IgG ($n = 38$, $N = 5$), Healthy + NMDAR-Ab ($n = 68$, $N = 10$), or PSY + NMDAR-Ab ($n = 131$, $N = 15$). Data are expressed as mean ± SEM. One out of one Healthy −, three out of three Healthy +, and four out of nine PSY + were used and pooled for comparisons. $P > 0.05$, Krukal–Wallis test

from Healthy −, Healthy +, or PSY + subjects. The size of synaptic EphB2R clusters was reduced by PSY + NMDAR-Ab when compared with Healthy + NMDAR-Ab (Fig. 6a, b). The number of synaptic EphB2R clusters was also decreased in the presence of PSY + NMDAR-Ab (Fig. 6c). We then performed single QD tracking to precisely assess EphB2R dynamics in synapses exposed to either NMDAR-Ab from Healthy + subjects or PSY + patients (Fig. 6d). In the presence of PSY + NMDAR-Ab, the distributions of EphB2R surface dynamics was significantly shifted (Fig. 6e) and the diffusion coefficient was increased. Thus, our data indicate that NMDAR-Ab from PSY + patients disturb EphB2R synaptic retention and content, suggesting that the loss of this anchoring partner of NMDAR could possibly contribute to the loss of synaptic NMDAR.

**NMDAR-Ab impair LTP at glutamatergic synapses**. NMDAR-Ab from PSY + patients decrease NMDAR synaptic content, likely impacting NMDAR-dependent processes such as synaptic long-term plasticity. To address this question, we first tested the direct and acute effect of NMDAR-Ab on the receptor function by monitoring spontaneous calcium transients mediated by the activation of NMDAR in live hippocampal networks. For this, the genetically encoded calcium indicator GCaMP3 was expressed in hippocampal neurons and calcium transients were recorded in

spines in the presence of an L-type voltage-dependent calcium channel blocker. The calcium transients were mediated by NMDAR, as the NMDAR-competitive antagonist D-AP5 (50 μM) fully abolished the events (Fig. 7a). A 5 min incubation of neurons with purified IgG from a Healthy − subject, NMDAR-Ab from Healthy +, or PSY + individuals altered neither the event frequency (Fig. 7b) and area (mean ± SEM, No IgG = 127.8 ± 10.05 au ms⁻¹, $n = 33$ spines, $N = 4$ neurons; Healthy − = 130.3 ± 8.65 au ms⁻¹, $n = 46$, $N = 5$; Healthy + = 131.5 ± 22.64 au ms⁻¹, $n = 67$, $N = 10$; PSY + = 120.5 ± 7.99 au ms⁻¹, $n = 131$, $N = 16$; $P = 0.1737$, one-way analysis of variance), nor the rise and decay times (data not shown) of NMDAR-mediated calcium events compared to the control condition (i.e., with no IgG), indicating that NMDAR-Ab do not act as acute modulators of the NMDAR channel activity.

To assess the functional impact of NMDAR-Ab on the glutamatergic synapse, we monitored the synaptic content of surface GluA1-AMPAR in basal conditions and after an activity-induced synaptic AMPAR potentiation, i.e., chemical LTP (cLTP) [36, 51]. Neurons were exposed to Healthy − (one subject), Healthy + subjects (three out of three subjects), or PSY + patients (four out of nine patients) NMDAR-Abs for 12 h, to alter the NMDAR synaptic content. The basal level of GluA1-AMPAR within synapses was significantly lowered by NMDAR-Ab from PSY + patients when compared with control or Healthy − conditions (Fig. 8a). After cLTP stimulation, a synaptic recruitment of AMPAR was observed, as evidenced by the progressive increase of GluA1-AMPAR cluster area (Fig. 8b) and intensity (Supplementary Fig. 6) over time at synapses exposed to NMDAR-Ab from Healthy + subjects and in control condition. Strikingly, cLTP-induced potentiation of GluA1-AMPAR was not achieved after exposure to NMDAR-Ab from PSY + patients (Fig. 8b and Supplementary Fig. 6), even leading to a tendency toward a depression of the AMPAR synaptic content. Thus, NMDAR-Ab from PSY + patients alter the basal level of synaptic AMPAR and impair their recruitment during activity-dependent synaptic plasticity. To further strengthen these in vitro observations, we performed a series of intra-hippocampal injections of NMDAR-Ab (Fig. 8c) in young rats (P12-P15), followed by patch-clamp recordings at CA3-CA1 synapses. LTP was induced by a high-frequency stimulation (HFS) protocol and excitatory postsynaptic currents (EPSC) were recorded at − 60 mV (Fig. 8d). Strikingly, PSY + NMDAR-Ab prevented NMDAR-dependent LTP, even inducing long-term depression, at these synapses (Fig. 8e, f). Together, our in vitro and ex vivo data converge upon the conclusion that NMDAR-Ab from PSY + patients specifically alter LTP at the hippocampal synapses.

## Discussion

In this study, we used a combination of classical cell-based assays and cutting-edge single-molecule imaging approaches to firmly identify circulating NMDAR-Ab in about 3% of healthy subjects and 19% of psychotic patients diagnosed with SCZ (acute phase). More importantly, we took further advantage of the single-molecule imaging to provide the evidence that NMDAR-Ab from PSY + patients, but not from healthy subjects, destabilize synaptic NMDAR and its interacting partner EphB2R. Consequently, NMDAR-Ab from PSY + patients decreased the NMDAR synaptic content over time and impaired NMDAR-dependent synaptic LTP. Noteworthy, NMDAR-Ab from the different PSY + patients showed a remarkable homogeneity in terms of molecular disturbance. Thus, although investigation of the in vivo impact of NMDAR-Ab from large cohorts of psychotic patients will be necessary, our evidence fuels the hypothesis that NMDAR-

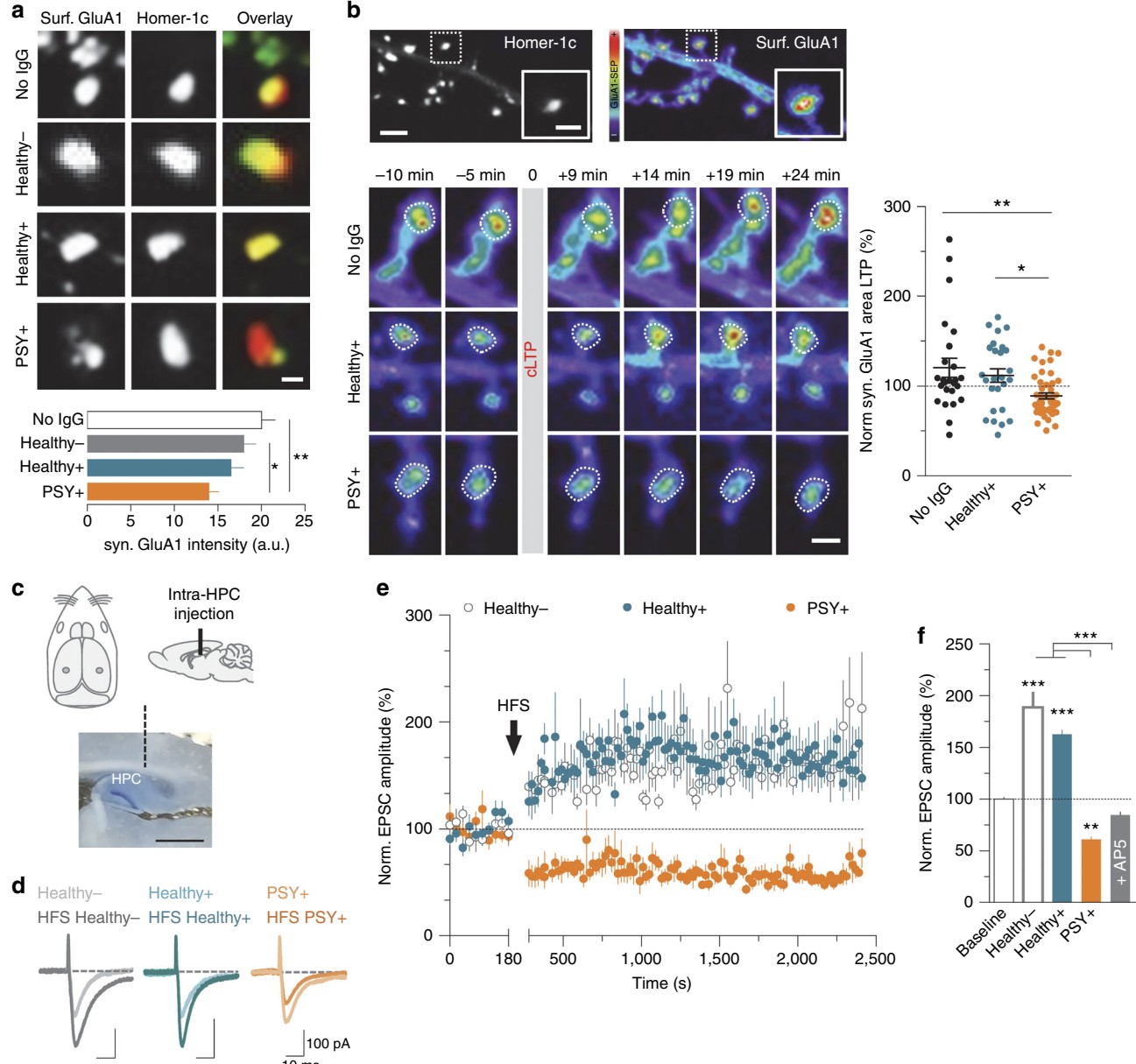

**Fig. 8** PSY + NMDAR-Ab prevent LTP expression through an alteration of basal and activity-dependent recruitment of synaptic AMPAR. **a** Live immunostaining of surface GluA1-SEP clusters (12 div) in control condition (No IgG) or after a 12 h incubation with purified IgG (5 µg ml⁻¹) from Healthy −, Healthy +, or PSY + subjects. Scale bar, 500 nm. Bottom panel: mean ± SEM of synaptic GluA1-AMPAR fluorescence intensity in control condition with no IgG ($n = 74$ dendritic regions, $N = 24$ neurons) or after exposure to Healthy − ($n = 41$, $N = 10$), Healthy + ($n = 52$, $N = 16$), or PSY + NMDAR-Ab ($n = 74$, $N = 20$). **$P = 0.0042$, Kruskal–Wallis test. **b** Impact of NMDAR-Ab on chemically induced long-term potentiation (cLTP) of synaptic AMPAR. The pseudocolor representation codes for GluA1-SEP fluorescence intensity levels. Scale bar, 20 µm; scale bar inset, 1 µm. Bottom panel: time course of GluA1-SEP fluorescence in a synaptic area (white dashed line) after a cLTP stimulus with Healthy + or PSY + NMDAR-Ab (5 µg ml⁻¹). Scale bar, 1 µm. Right panel: normalized synaptic GluA1-AMPAR clusters area 19–24 min after application of the cLTP stimulus with no IgG ($n = 25$ neurons), Healthy + ($n = 26$), or PSY + NMDAR-Ab ($n = 46$). Data are expressed as mean ± SEM. **$P = 0.016$, one-way ANOVA. **c** Schematic description of the stereotaxic injection of purified IgG (5 µg ml⁻¹) into the dorsal hippocampus of P12–P15 rats. Scale bar, 1 mm. **d** Representative EPSC traces recorded during baseline or 35–40 min after the high-frequency stimulation (HFS) protocol following injection of purified IgG from Healthy − (one out of one subject), Healthy + (pool of three out of three samples), and PSY + (pool three out of nine samples) individuals. **e** Average time course of HFS-induced LTP. Normalized EPSC amplitudes (normalization to the mean amplitude of EPSCs recorded during baseline acquisition) are plotted against time for Healthy − ($n = 8$ neurons), Healthy + ($n = 10$), and PSY + ($n = 9$) conditions. Data are presented as mean ± SEM. **f** Mean ± SEM of normalized EPSC amplitudes at baseline or 35–40 min after application of the pairing protocol in Healthy − ($n = 8$), Healthy + ($n = 10$), PSY + ($n = 9$), or D-AP5 ($n = 5$) conditions. ***$P < 0.0001$; one-way ANOVA

Ab from PSY + patients contribute to the NMDAR hypofunction underlying behavioural deficits in animal models of psychosis.

Over the last years, several studies have reached different conclusions regarding the presence of NMDAR-Ab in psychiatric disorders[2, 4, 25]. Although there is no straightforward explanation

for these discrepancies, the different methods used to define sample seropositivity have been at the centre of debates. In order to optimize the autoantibody detection beyond classical cell-based assays (using either fixed or live HEK293 cells expressing ectopic receptors), we implemented a single molecule tracking approach

to define the live cellular signature of the autoantibody endogenous membrane target. Although such imaging approach is technically and currently challenging to implement as a clinical diagnostic[52], we could demonstrate that the cellular signature of NMDAR-Ab is undistinguishable from the one of native GluN1-NMDAR. The seropositivity of psychotic patients diagnosed with SCZ also raises the questions of whether they suffered from anti-NMDAR autoimmune encephalitis. A thorough clinical examination of our seropositive patients does not support this claim (see Results). Ongoing and future large clinical studies will surely shed key lights on that issue that seropositive patients possibly constitute a distinct group that share psychotic symptoms with SCZ, referred to as "autoimmune psychosis". Noteworthy, patients with anti-NMDAR encephalitis do not appear to develop psychotic disorders despite a remaining low level of NMDAR-Ab in sera and even CSF for months to years in few cases[26, 53]. This suggests that other unknown processes are likely involved in the etiology of psychotic disorders. Genetically based immune or synaptic instability, developmental stage(s) at which autoantibodies are present in the brain, or altered biological barriers are potential candidates. In addition, whether NMDAR-Ab from PSY + and encephalitis patients act differently on their target remains to be further examined, as we unveiled both clear differences (lower NMDAR-Ab titer in PSY +, non-overlapping binding on NMDAR, absence of NMDAR-Ab in CSF) and similarities (alteration of NMDAR surface diffusion and synaptic content, alteration of EphB2R trafficking, blockade of synaptic AMPAR potentiation, no effect on the NMDAR-mediated calcium influx) between both disorders. The NMDAR signaling hypofunction is increasingly accepted as part of the etiopathology of psychosis and SCZ, as the discovery that certain NMDAR blockers reproduce a large spectrum of SCZ-like symptoms[30]. In addition, alterations of NMDAR synaptic content have consistently been reported in the brains of schizophrenic patients[54, 55]. In line with this, we here report that the presence of NMDAR-Ab from PSY + patients, but not Healthy + subjects, strongly decrease NMDAR synaptic content and disturb its synaptic nanoscale organization. Noteworthy, NMDAR-Ab detected in patient with autism spectrum disorder, but without any history of psychosis, were not able to perturb NMDAR synaptic trafficking[56].

Abnormal NMDAR surface distribution and/or activity can contribute to the emergence of neurological and psychiatric disorders[33]. Surface trafficking represents a key mechanism to control NMDAR synaptic localization and regulate NMDAR-mediated synaptic plasticity, which contributes to the maintenance of many physiological functions such as memory. Indeed, by using anti-NMDAR antibody to alter the receptor surface diffusion it was demonstrated that the basal surface dynamics has a key role in the establishment of synaptic plasticity[36] and hippocampal-associated memory processes[43]. Of great interest, human autoantibodies against NMDAR from encephalitis patients have the similar potency to perturb the NMDAR surface trafficking and synaptic content[32, 35], to abolish synaptic plasticity processes and to induce behavioral deficits[32, 36, 37, 45, 57, 58], without altering NMDAR-dependent calcium activity[32, 36, 59]. Our findings that only NMDAR-Ab from PSY + patients, and not Healthy +, have the potency to dynamically disorganize synaptic NMDAR and alter NMDAR-dependent network processes further strengthen the idea that profound alterations of NMDAR surface trafficking might contribute to the emergence of psychotic symptoms through a NMDAR hypofunction. This should pave the way for the development of new therapeutical strategies to successfully treat PSY + patients, favoring immune therapies and/or molecules that balance NMDAR trafficking rather than pharmacological modulations of the channel activity. Of interest,

immunotherapy treatment of seropositive patients with acute psychosis has been associated with good outcomes, especially for psychosis resistant to anti-psychotics[28, 29]. Moreover, D-serine, a co-agonist of the NMDAR that strongly alter the receptor surface dynamics[60], clinically improves the symptoms of a NMDAR-Ab PSY + patient[61].

In conclusion, super-resolution imaging approaches were implemented in hippocampal cell networks to provide the surface organization map of NMDAR at the nanoscale level. In basal condition or in presence of NMDAR-Ab from healthy subjects, NMDAR laterally diffuse and stabilize in nanometer-sized clusters within glutamatergic synapses. We could unmask a specific alteration of the NMDAR synaptic complex and trafficking by NMDAR-Ab from PSY + patients. As NMDAR-Ab do not alter the NMDAR calcium influx, our data support a model in which the NMDAR hypofunction that has been linked to psychosis is induced by preventing the receptor synaptic anchoring and subsequent altered trafficking. Future investigations unveiling the molecular cascade underlying NMDAR destabilization (e.g. EphB2R interaction, or intracellular signalling cascade) and identifying NMDAR-Ab epitope(s) will be of prime importance. In a clinical perspective, we demonstrated that circulating NMDAR-Ab can have distinct molecular impact, and thus their mere detection is not sufficient to predict a NMDAR signaling dysfunction.

## Methods

**Participants**. Schizophrenic patients ($n = 48$) meeting DSM-IV criteria (APA, 1994) were included in this study after approval by a French ethical committee (Comités de protection des personnes, Comité consultatif sur le traitement de l'information en matière de recherche, Commission nationale de l'informatique et des libertés) and written informed consent for their participation. Patients were interviewed with the French version of the "Diagnostic Interview for Genetic Studies" for the assessment of lifetime clinical characteristics of SCZ. Positive and negative symptoms were assessed with the PANSS and cognitive function with the National Adult Reading Test (NART). In addition, medical history was explored, in particular history of neurological or inflammatory disorders. Healthy subjects ($n = 104$) without any personal and family history of SCZ or bipolar disorder were enrolled through a clinical investigation center (Center for Biological Resources, Mondor Hospital, France). Participants were included only if they were negative for HIV1/2, Hepatitis B and C, had no ongoing inflammatory, auto-immune or neurological disorders, and no ongoing immunosuppressive or immune-modulating treatment.

**Collection of blood samples and medical examination**. Blood samples were collected from patient and control groups within 1 week of the clinical assessment. Sera were then purified in order to extract IgG isotype antibodies. Samples were dialyzed against phosphate-buffered saline (PBS) and solutions were used at pH 7.4[32, 62]. Seropositive patients for NMDAR-Ab were contacted for additional medical and biological evaluations. The remaining volume of sera after infectious screening and NMDAR-Ab detection (see below) was however not sufficient in each patient to perform IgG purification (four of nine were then used for purified IgG-based assays).

**NMDAR-Ab detection in heterologous cells and neuronal preparations**. Serum samples were tested for the presence of NMDAR-Ab using a cell-based assay on human embryonic kidney cells (HEK293) expressing both GluN1 and GluN2B subunits of the NMDAR, using the method previously described[32, 62]. To detect NMDAR-Ab in CSF, cells were fixed (4% paraformaldehyde, 10 min) and then incubated with patients' CSF (1:0 in saturation buffer, 90 min). Samples were considered as positive when a clear staining was confirmed by three different readers in three independent assays. Titers of positive sera were determined by end point dilutions. For cellular imaging processes (for further details, see Supplemental Experimental Procedures), either sera or purified IgG from seropositive subjects were used. Cultures of hippocampal neurons were prepared from E18 Sprague–Dawley rats and were maintained at 37 °C in 5% $CO_2$ for 15 div at maximum as previously described[32]. For exogenous protein expression, 7–10 div hippocampal cultured neurons were transfected at least 48 h before each experiment using either the Effectene (Qiagen) or phosphate calcium transfection.

**Immuno-absorption**. Schematically, 24 h after plating (150,000 cells per ml), HEK293 cells were transfected with GluN1-SEP and GluN2B (Lipofectamine LTX, Invitrogen). The following day, either transfected or non-transfected HEK cells

were incubated for 1 h with NMDAR-Ab (5 µg ml$^{-1}$). This step was repeated 6 times and the resulting absorbed fraction was kept at 4 °C (max 24 h) for further live experiments.

**Immunohistochemistry.** Mice (P22) hippocampal coronal sections of 50 µm were incubated overnight at 4 °C with either a polyclonal antibody against the N-terminal part of the GluN1 subunit (αGluN1$_{N-term}$ Alomone Labs, 20 µg ml$^{-1}$) or human purified IgG (20 µg ml$^{-1}$). Fluorescent revelation was carried out with secondary anti-rabbit or anti-human Alexa 488 antibodies (Life Technologies, 1/1,000) for 2 h at room temperature. Images were obtained using a Nanozoomer and a confocal microscope (SP8, Leica). Brains (provided by A. Ramsey) from wild-type and GluN1-knockdown animals at 14 weeks were perfused and stored at − 20 °C. Coronal tissue sections of hippocampal areas (20 µm thick) were cut on a micro-tome-cryostat, thaw-mounted onto Thermo Scientific, SuperFrost Ultra Plus adhesion slides and stored at − 20 °C until further processing. Sections were fixed at 4 °C in 4% paraformaldehyde. Blocking was carried out in a 1 × TBS solution in 0.3 M glycine containing 10% normal goat serum (Sigma) and incubation with PSY + purified IgG (5 µg ml$^{-1}$, pooled from two different patients) was done in a 1 × TBS solution containing 10% normal goat serum overnight at 4 °C. Staining with secondary antibody anti-human Alexa 568 (Invitrogen, 1/500) was performed for 1 h and slides were mounted with Vectashield + Dapi (Vector Laboratories). Image acquisition was done on a video spinning-disk system (Leica DMI6000B, × 40).

**Immunocytochemistry.** Human IgG labeling was obtained on fixed dissociated hippocampal neurons after incubation with purified IgG (5 µg ml$^{-1}$, overnight 4 °C) and a secondary anti-human Alexa 488 antibody (1/500, 30 min). Hippocampal cultured neurons expressing surface exogenous GluN1-SEP NMDAR or GluA1-SEP AMPAR were stained using a monoclonal anti-GFP antibody (1/500, 20 min), surface endogenous EphB2R were labeled with an anti-EphB2R polyclonal antibody (1/200, 3 h) followed by Alexa 488-conjugated anti-mouse or anti-goat secondary antibodies (1/500, 30 min). Synaptic labelling was obtained using an anti-Homer-1c antibody (1/500, 30 min) and an Alexa 594-conjugated anti-guinea pig secondary antibody (1/500, 30 min). All imaging sessions were done on a videospinning-disk system (Leica DMI6000B, × 63) and quantification analysis was performed using MetaMorph software (Molecular Devices) and ImageJ (NIH).

**Immunocompetition.** Sections (7 µm) of hippocampal tissue were fixed and incubated with undiluted serum from patients or blood donors overnight at 4 °C. Sections were then extensively washed with cold PBS and incubated for 1 h with biotinylated IgG from a representative patient with NMDAR-Ab. After washing, the binding of biotinylated IgG was revealed with a standard avidin–biotin–peroxidase method (Vectastain ABC kit Elite, PK-6100, Vector). Slides were then mildly counterstained with hematoxylin, mounted, and results photographed with a digital camera (AxioCam MRc) adapted to a confocal microscope (Zeiss LSM710). Hippocampal cultured neurons were fixed and successively incubated with PSY + NMDAR-Ab (5 µg ml$^{-1}$, overnight, 4 °C). Remaining antigen binding sites were blocked using anti-human Fab fragments (Jackson ImmunoResearch, 100 µg ml$^{-1}$, 1 h). Cells were then incubated with competing IgG from PSY +, Healthy +, or Enceph + individuals (5 µg ml$^{-1}$, overnight, 4 °C). All imaging sessions were done on a video spinning-disk system (Leica DMI6000B, × 63) and quantification analysis was performed using ImageJ (NIH).

**Direct stochastic optical reconstruction microscopy.** Live hippocampal neurons were incubated with Healthy + or PSY + NMDAR-Ab for 2 h at 37 °C. Surface endogenous GluN2A-containing NMDAR were specifically stained using an anti-GluN2A antibody (0.1 mg ml$^{-1}$, 15 min). Cells were then successively incubated with an anti-PSD95 antibody (0.1 mg ml$^{-1}$, 45 min), secondary anti-rabbit Alexa 647 (Invitrogen, 0.1 mg ml$^{-1}$, 30 min), and anti-mouse Alexa 532 (0.1 mg ml$^{-1}$, 30 min) antibodies. A second fixation was performed after incubation with the secondary antibodies. All imaging sessions were performed using a Leica SR GSD 3D microscope (Leica HC PL APO 160 × 1.43 numerical aperture oil-immersion total internal reflection fluorescence objective) and an ANDOR EMCCD iXon camera. Localization of single molecules and reconstruction of the super-resolved image was performed by applying a fitting algorithm determining the centroid-coordinates of a single molecule and fitting the point-spread-function of a distinct diffraction limited event to a Gaussian function. The final achieved spatial resolution was 40 nm.

**QD tracking and surface diffusion calculation.** QD labeling and microscopy were performed as previously described[32]. See Supplemental Experimental Procedures for further details.

**Calcium imaging.** Dissociated neurons transfected with GCaMP$_3$ at 9–10 div were transferred into a Tyrode solution at 13–15 div. Fifteen minutes before imaging, cells were transferred to a Mg$^{2+}$-free Tyrode solution with 5 µM Nifedipine (Tocris) and 5 µM Bicuculline (Tocris). Time-lapse images were acquired at 20 Hz.

Three time-lapse movies (3,000 frames) were successively recorded: (1) "Pre" (baseline period), (2) "Post" (5 min after bath application of buffer or purified IgG, 5 µg ml$^{-1}$), and (3) "D-AP5" (5 min after bath application, 50 µM).

**Chemically induced potentiation (cLTP).** Live hippocampal neurons (12 div) transfected with GluA1-SEP at 10 div were incubated overnight with Healthy + or PSY + NMDAR-Ab (5 µg ml$^{-1}$, 37 °C). After washing thoroughly, chemically induced cLTP was elicited by a bath co-application of glycine (200 µM) and picrotoxin (5 µM) for 4 min. cLTP was always applied after a period of baseline acquisition and the medium was carefully replaced by fresh equilibrated and heated medium after induction. GluA1-SEP fluorescence signal was then recorded every 5 min during the 30 min following the stimulus. Synapses were defined using the synaptic protein Homer-1c DsRed. Synaptic GluA1-SEP clusters intensity and area values were normalized to the baseline values.

**In vivo hippocampal injections and electrophysiology.** Briefly, P12–15 Sprague–Dawley rats were anesthetized by inhalation of isoflurane. One microliter of purified IgG (5 µg ml$^{-1}$, mix of several samples) from one out of one Healthy −, two out of three Healthy +, or three out of nine PSY + individuals were infused into the dorsal hippocampus and the stereotaxic coordinates were adapted according to the age of the animals (coordinates relative to bregma, from AP: − 3.5 mm, ML: ± 2 mm, DV: − 2 mm at P12, to AP: − 3.8 mm, ML: ± 2 mm, DV: − 2.5 mm at P15). Two to 3 h after intra-hippocampal injections, P12–15 Sprague–Dawley rats were anesthetized and sagittal brain slices (350 µm thick) were prepared in an ice-cold sucrose buffer solution containing (in mM): 250 sucrose, 2 KCl, 7 MgCl$_2$, 0.5 CaCl$_2$, 1.15 NaH$_2$PO$_4$, 11 glucose, and 26 NaHCO$_3$ (gassed with 95% O$_2$/5% CO$_2$). Slices were then incubated for 30 min at 33 °C and subsequently stored at room temperature in an artificial CSF (ACSF) solution containing (in mM): 126 NaCl, 3.5 KCl, 2 CaCl$_2$, 1.3 MgCl$_2$, 1.2 NaH$_2$PO$_4$, 25 NaHCO$_3$, and 12.1 glucose (gassed with 95% O$_2$/5% CO$_2$; pH 7.35). Electrodes (4–5 MΩ) were filled with a solution containing (in mM): 120 cesium methanesulfonate, 4 NaCl, 4 MgCl$_2$, 10 HEPES, 0.2 EGTA, 4 Na$_2$ATP, 0.33 Na$_3$GTP, and 5 phosphocreatine adjusted to pH 7.3 with CsOH. EPSCs were evoked at a rate of 0.05 Hz using an ACSF-filled glass micropipette positioned in the stratum radiatum to stimulate Schaffer collaterals. Currents were recorded at − 60 mV in the presence of the GABA$_A$ and GABA$_B$ receptor blockers SR95531 (10 µM) and CGP55845 (5 µM), respectively. Under these conditions and after a stable eEPSC recording had been maintained for 10 min, tetanic stimulation (4 trains of 100 stimuli at 100 Hz, delivered at 20 s interval) of Schaffer collaterals was used to induce LTP. The access resistance was monitored throughout the experiment and data were discarded when it changed by > 20%.

**Data comparison and statistics.** All values and statistical comparisons are detailed in Supplementary Table 4.

**Data availability.** The data that support the findings of this study are available from the corresponding author upon reasonable request.

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

## Acknowledgements

This work was supported by the Centre National de la Recherche Scientifique, Agence Nationale de la Recherche (ANR-12-SAMA-0014), Fondation pour la Recherche Médicale, Conseil Régional d'Aquitaine, Labex Bordeaux BRAIN, IDEX Bordeaux, fondation FondaMental, Labex Bio-PSY, and Ministère de l'Enseignement supérieur et de la

Recherche. We thank the Bordeaux Imaging Center (service unit of the CNRS-INSERM and Bordeaux University, member of the national infrastructure France BioImaging) for confocal and STORM imaging, and J.B. Sibarita for STORM analysis. We thank Charlotte Bertot and Pauline Durand for technical assistance on cell cultures, molecular biology, and immunohistochemistry, Daniel Jercog for calcium imaging analysis, and lab members for constructive discussions.

## Author contributions

J.J., E.M.J., J.H., M.L., and L.G. designed the experiments. J.J., E.M.J., J.D., V.R., M.S., J.D., J.H., M.L., and L.G. designed the methodology. E.L., C.R., N.H., R.H., J.H., and M.L. performed clinical analysis. J.J., E.M.J., J.D., H.G., V.R., M.S., M.L., B.K., N.H., E.L., C.R., E.M., R.H.Y. performed the experiments. J.J., E.M.J., J.D., H.G., V.R., M.S., M.L., B.K., and E.M. analyzed the data. A.J.R. provided material. J.J., E.M.J., J.D., J.H., M.L., and L.G. wrote the paper.

## Additional information

**Competing interests:** The authors declare no competing financial interests.

