## [Peer Review File · Nature Communications]

Reviewers' comments:

Reviewer #1 (Remarks to the Author):

In this study, Jézéquel et al., describe the identification of NMDA receptor autoantibodies in serum from healthy and psychotic patients. Using single-molecule imaging approaches they provide evidence that these circulating autoantibodies from psychotic, but not healthy individuals alter the dynamics of NMDARs on the surface of neurons. Moreover, surface staining experiments on neurons revealed that on the long term (>12 hours) autoantibodies from psychotic patients reduce the surface expression of NMDARs, but also EphrinB2 and AMPA receptors. Functionally, these autoantibodies do not seem to acutely alter the calcium influx through NMDARs, but neurons exposed to these autoantibodies for longer durations are unable to upregulate synaptic AMPARs in response to a chemically-induced synaptic potentiation protocol. Overall, this study is well-performed, presents the data in clear and visually attractive images, includes rigorous statistical analyses and certainly has potentially important implications for understanding the etiology of psychosis. The finding that autoantibodies against the NMDAR can be detected in both healthy and psychotic individuals with differential effects on NMDAR functioning is a novel and important finding for the field. However, some of the conclusions are not convincingly supported by the experimental data as it is presented now, and additional evidence is required to demonstrate conclusively that the autoantibodies from psychotic patients have different effects than the autoantibodies from healthy subjects.

Major comments

- The authors present several tests to demonstrate the ability of autoantibodies to detect NMDARs, but it is unclear how specific these autoantibodies are for NMDARs. Especially the images in Figure 1d are very hard to interpret, there is clear staining with the human IgGs visible in GFP-negative areas. Also, the other way around, in the PSY+ group, a clear GFP-positive area is IgG-negative. The degree of specificity of the autoantibodies for the NMDAR has important implications for the interpretation of the results, and the authors should address this by doing at least some of the following controls:

- (1) The authors should include (images of) appropriate negative controls: human IgG staining on untransfected HEK cells, and/or human IgG staining on HEK cells transfected with another GFP-tagged synaptic surface protein.
- (2) The authors could include a staining with NMDAR-Ab negative serum from healthy individuals on GluN1/2 transfected HEK cells.
- (3) The authors should correlate the staining intensity with the GluN1/2 expression level. A clear, positive correlation between GFP intensity and human IgG staining intensity would very much strengthen their point.
- (4) Figure 1c shows some, but not convincing co-localization of the autoantibodies with Homer-1c. Co-localization with a commercial or commonly accepted GluN1 antibody would more directly demonstrate the specificity of the autoantibodies.
- (5) The authors should consider to re-do the absorption assay as in Figure 3g, and test whether staining with the autoantibodies is lost after pre-absorption on GluN1/2 transfected

HEK cells (also see below).

(6) Ideally, the authors would provide data that IgG staining is absent in tissue or neurons from GluN1 knockout mice, or reduced in GluN1 knockdown neurons.

- The experiments presented in Figure 3 lack the comparison with a proper negative control, e.g. untreated cells or cells treated with an unrelated human IgG. This is in fact also the case for experiments presented in Figure 4 and 6, see below. The authors should include this, without this control it is difficult to interpret the differential effect of the Healthy+ and PSY+ derived autoantibodies on NDMAR diffusion.

- The experiment presented in Figure 3g is a very important and elegant approach to test the specificity of the autoantibodies (but on itself not sufficient). However, the authors describe that they do the pre-absorption on GluN1-transfected cells, while they describe earlier that these autoantibodies did not stain HEK cells that only express GluN1, suggesting that these do not recognize GluN1 in isolation, only in the presence of GluN2 subunits. The authors should address this contradiction.

- Figure 4 presents super-resolution images of GluN2A-containing NMDARs and reveals 'nano-objects' that are clustered together in groups of 4-6. This is a very intriguing finding, and as the authors note, one of the first investigations into the nanoscale organization of NMDARs. However, data for untreated control cells is lacking entirely. This should be included for comparison with the experimental conditions. Perhaps it would also be interesting to compare measures of domain organization between extrasynaptic and synaptic clusters. Also, could the authors elaborate a little bit more on the function of these domains? Perhaps the authors could discuss this finding in the light of other comparable studies describing these domains for other synaptic proteins (e.g. Nair et al., 2013, MacGillavry et al., 2013, Specht et al., 2013).

- Figure 6 presents data that EphB2R levels are reduced by autoantibodies from psychotic patients. However, also in neurons treated with autoantibodies from the healthy group, only 40% of the synapses are EphB2R positive. Is this a normal fraction found also in control neurons, or do the antibodies from the healthy+ group also affect EphB2R levels? The authors should present data for a proper negative control group.

- The authors claim that "...our data demonstrate that NMDAR-Ab from PSY+ patients specifically disturb EphB2R synaptic retention and content, suggesting that the loss of this anchoring partner of NMDAR is likely responsible for the destabilization of the receptor in presence of PSY+ NMDAR-Ab." The evidence for their conclusion that the autoantibodies disrupt the interaction between synaptic EphrinB2 receptors and thereby uncouple NMDARs, however is very weak. The only piece of evidence that is provided is that the synaptic levels of both NMDARs and EphB2Rs are reduced after chronic exposure to the autoantibodies. This is in line with their previous work, but on itself not more than an interesting correlation. The authors should provide more compelling evidence that the autoantibodies disrupt the interaction between EphB2R and NMDARs, or adjust their conclusion.

Also, the conclusion that this is specific for EphB2R is too far of a stretch, as they for

instance also found a reduction in AMPAR levels, seemingly indicating a more general effect on synaptic receptor levels. In that respect, alternative explanations are hardly explored or discussed. For instance, perhaps the most straight-forward explanation for reduced surface expression is that IgG binding to NMDARs induces their internalization (along with interactors such as EphB2R). This should be addressed experimentally, or at least be discussed as an alternative explanation for their results.

Minor comments

- The authors present their single-molecule tracking approach as an unambiguous assay to test the specificity of the human IgGs. Even further, in the discussion the authors propose that their single-molecule approach could be a diagnostic standard to detect NMDAR-Abs in human serum. They do point out that this is technically challenging, but besides this and other practical considerations (e.g. experimental reproducibility), at best, this approach will only provide evidence that the autoantibody binds an epitope on a protein that has similar mobility characteristics as the NMDAR: it cannot provide unambiguous evidence that the IgG at test is an NMDAR-specific Ab. The authors should re-phrase or remove this statement from the manuscript.

- The authors conclude that "...only patients' NMDAR-Ab prevent chemically-induced AMPA receptor synaptic potentiation while leaving intact NMDAR-mediated calcium influx". However, these two outcomes (intact calcium influx, but no potentiation) are measured at completely different time scales, after 5 minute vs. 12 hr incubation times. After 12 hr incubation with the PSY+ autoantibodies NMDAR levels were severely reduced, and it is very likely that at that point NMDA-mediated calcium influx is also disrupted. Thus, it is unknown whether the effect on plasticity is independent of the effect on NMDAR-dependent calcium signaling. These two conclusions should be more clearly separated, or it should be tested experimentally that calcium influx is not disrupted after 12 hr incubation with IgGs.

- To me the determination of the autoantibody titers is confusing. It is described that "titers were estimated as end point dilutions". Could the authors explain this in a little bit more detail? What is the unit in Figure 1b? The term 'CBA' is also used in the discussion (line 330), but not explained.

- In Figure 8d, the authors present a quantification of GluA1 cluster area. Presenting intensity data would be more appropriate here as an increase in AMPAR levels is expected, not necessarily synaptic area. Also, with respect to the statistical analysis, it seems that in 8b and 8d the IgG groups are each compared with the control group. Is the PSY+ group also significantly different from the Healthy+ group?

Reviewer #2 (Remarks to the Author):

The present manuscript by Jézéquel et al. is an interesting study examining the possible mechanism of action of circulating autoantibodies against the NMDARs in patients with

schizophrenia. This is a timely and important topic given the significant level research on anti-receptor antibodies in neuropsychiatric disorders since their discovery in idiopathic encephalitis. Because of early expression of schizophrenia-like psychotic symptoms in patients with anti-NMDAR antibody-associated encephalitis, there has been significant interest in determining a potential role in a subset of patients with schizophrenia without a history of overt encephalitis. This manuscript is novel in that it uses anti-NMDAR antibodies from patients diagnosed with schizophrenia but without an overt history of encephalitis.

The authors find anti-NMDAR antibodies in just under 20% of schizophrenia patients tested, consistent with the higher end of previous studies, and interestingly 3% of a non-schizophrenia, non-psychiatric "health control" cohort. A strength of this study is the direct comparison and differential effects between the anti-NMDAR antibodies from schizophrenia patients and healthy controls suggesting the specific location of the antibody isotope may dictate the expressed symptoms.

Their major novel finding is that anti-NMDAR antibodies from patients with schizophrenia, but not those from healthy controls, alter the synaptic mobility and organization of NMDARs and possibly the associated EphrinB2 receptor. They go on to suggest that this disorganization leads to impaired synaptic plasticity. The experiments are well-executed and the data are logically presented. However, there are some concerns that need addressed.

Major issues:

1) It is very much unclear to me when the authors used pooled human antibodies and when they tested each sample independently. This is of critical importance for the conclusions the authors are making. Do every sample from patients cause the synaptic NMDAR disruption and does every sample from controls not cause the disruption? If pooled, the effects could be induced by a single dominant sample, and the ability to make any sort of generalization is lost. If individual samples were used, the presented data in each figure needs to represent the variation between samples rather than, for example, variation between regions of interest in the imaging. This is a crucial distinction.

2) It was noted that in no cases were anti-NMDAR antibodies detected in the patients' CSF. If circulating CSF antibodies are not found in the brain, this limits the clinical utility of these findings. The authors, however, tested for CSF antibodies on fixed HEK cells whereas the earlier said with serum samples that fixed HEK cells did not work for detection and they thus used live HEK cells. Why the difference? Also, how does the concentration of Ab used in these experiments relate to the Ab exposure of synaptic NMDARs, especially if there is none in the CSF?

3) For many figures it would be useful to have additional controls. Figure 1 – positive controls with commercial antibodies, negative controls of purified IgG from negative subjects, negative controls from untransfected HEK cells, etc. For other experiments, purified human IgG from negative subjects as a negative control would be useful. The distinction between the schizophrenia+ samples and the healthy+ samples is very interesting, but hard to know if the healthy+ sample are doing anything.

4) The reduction of EphB2R from synapses in Figure 6 is interesting, does this suggest a disruption of the interaction with NMDARs or perhaps they are moving together? Ideally, some evidence of a disruption of the interaction would be supportive, though at the minimum, additional discussion is warranted.

5) Figure 7 does not provide enough evidence to make the conclusion that anti-NMDAR antibodies do not effect Ca²⁺ transients. The frequency of the events says very little about the NMDAR responses if there is not a loss of synapses or complete loss of NMDARs at synapses. Analysis of the amplitude, area, and decay kinetics of the Ca²⁺ transients is needed at the minimum. Electrophysiological analysis in these samples would be additionally be welcome.

6) Figure 8 would likewise be more supported with some electrophysiological analysis.

7) Are these effects GluN2A-containing receptor specific, or are there also effects of GluN2B-containing receptors?

8) Why is the MSD in figure 1f so different from the MSD figure 3d?

Minor issues:

- Line 37 (abstract): should "liability" be "reliability"?
- Line 62: associated with psychotic symptoms
- Line 117: maybe "past" instead of "ancient"?
- Line 121: I think this refers to supplementary table 1 instead of 3
- Unsure why authors used "PSY+" instead of "SCZ+" to be consistent throughout manuscript
- Suppl Figure 1 could use some sort of quantification to say "comparative labeling was observed" on line 155.
- Line 167: please provide a reference to support this statement
- Figure 2e either needs much-improved labeling and/or should be moved to supplemental data
- Suppl Figure 3b: I don't see the commercial antibody line (black line in the key) on this graph
- The pre-absorption experiment was clever... not an issue, just wanted to say I liked it
- Suppl Figure 4b: how do you explain the reduction in synaptic GluA1 in Figure 8a without any changes in GluA1 movement? Perhaps add discussion.
- Figure 4a and Line 244: how do you know the surface GluN2A is "in glutamatergic synapses"?
- Lines 248-250: Sentences for Fig 4c and 4d seem swapped and should be in order of figure.
- Line 290: I don't like the use of "per se" here
- Line 312-313: "impair the plastic range of glutamatergic synapses" is overreaching.

Reviewer #3 (Remarks to the Author):

In this study, Jézéquel et al reported that autoantibodies against NMDAR (NMDAR-Ab) is differentially expressed in some portion of patients with schizophrenia (SZ) and healthy controls. The authors also demonstrated that the surface dynamics of synaptic NMDAR and EphrinB2, its anchoring receptor, is altered by treatment with NMDAR-Ab collected from patients with SZ, but not those from healthy controls using cutting-edge technologies, including single molecules imaging with Quantum dots and super-resolution microscopy. Quantum dot method was the technology that the authors previously used. In fact the authors previously demonstrated that NMDAR-Ab from patients with cognitive deficits prevent synaptic potentiation without affecting NMDAR function (Dupuis et al, EMBO Journal 2014). Thus the methods have been used by this group before and are not novel (though still cutting-edge). The results of the effect of NMDAR-Ab from patients with SZ are interesting, however, the sample size is too limited (3 seropositive healthy control and 4 seropositive SZ patients [though 9 SZ patients were identified as seropositive, only 4 is available for experiments]) to obtain definitive assessment. This reviewer is not convinced from the current data to see the significance to schizophrenia.

Reviewer #4 (Remarks to the Author):

This is an excellent study of autoantibodies to the NMDAR in psychosis. While the methods and results are very interesting, there are additional issues that should be addressed. Of great interest is the demonstration of functional differences in autoantibodies from subjects with and without psychosis. While the authors provide convincing evidence of differential effects of what appear to be the same autoantibodies. To further demonstrate this point, it would be critical to show that the autoantibodies from patients and controls are binding to different epitopes on the NMDAR. In the absence of these data, the authors should at least discuss the epitopic differences as an explanation for the differential autoantibody function between psychosis positive and negative antibody positive subjects. Further, if one wanted to design a diagnostic, as the authors propose, it would be best to use the epitope peptides as they would likely be pathologically significant based on the evidence presented in this manuscript.

We first would like to thank the four reviewers for their positive and constructive comments, which contribute to strengthen our manuscript.

Reviewer #1

We thank the reviewer for her/his positive and constructive comments, which strengthen our manuscript.

The authors present several tests to demonstrate the ability of autoantibodies to detect NMDARs, but it is unclear how specific these autoantibodies are for NMDARs. Especially the images in Figure 1d are very hard to interpret, there is clear staining with the human IgGs visible in GFP-negative areas. Also, the other way around, in the PSY+ group, a clear GFP-positive area is IgG-negative. The degree of specificity of the autoantibodies for the NMDAR has important implications for the interpretation of the results, and the authors should address this by doing at least some of the following controls: (1) The authors should include (images of) appropriate negative controls: human IgG staining on untransfected HEK cells, and/or human IgG staining on HEK cells transfected with another GFP-tagged synaptic surface protein.

Reply: We agree with the reviewer, as these controls are constantly performed for clinical detection of autoantibodies. We have thus added the results of these control experiments to the revised version of the manuscript (e.g. see Fig. 1 and Supplementary Fig 1). We performed immunostainings both on HEK cells and dissociated hippocampal neurons. First, we did not detect any staining on GluN1/GluN2B-transfected HEK cells after incubation with a seronegative sample (Healthy-) whereas a clear colocalization was observed in presence of a seropositive sample, comparable to the co-detection obtained in presence of a commercial anti-GluN1 antibody (Fig. 1d) or a positive serum from a patient with NMDAR encephalitis (Supplementary Fig. 1). In addition, neither seronegative nor seropositive samples colocalized with IgLON5-transfected cells (another surface protein involved in autoimmune encephalopathies), further confirming that autoantibodies present in the seropositive samples do target surface NMDAR. We reached the same conclusions using purified IgG on dissociated cultured hippocampal neurons. Indeed, there was no colocalization between neurons transfected with GluA1-SEP and the resulting staining from Healthy+ and PSY+ purified IgG. On the opposite, we could clearly detect overlapping signals between purified IgG from seropositive samples and GluN1-SEP expressing neurons, as well as with commercial anti-GluN1 antibody (Fig. 1). Of note, incubation of human serum on live neurons trigger cell death in a time-dependent manner, justifying our use of purified IgG throughout the entire study.

(2) The authors could include a staining with NMDAR-Ab negative serum from healthy individuals on GluN1/2 transfected HEK cells.

Reply: We have included such a control staining in the new version of the manuscript (see Fig. 1).

(3) The authors should correlate the staining intensity with the GluN1/2 expression level. A clear, positive correlation between GFP intensity and human IgG staining intensity would very much strengthen their point.

Reply: As the reviewer is well-aware, the fluorescence intensity of membrane receptor staining and GFP expression depend on several distinct parameters. For instance, the membrane staining will reflect the membrane content whereas the GFP will mostly reflect the protein expression (intracellular content). Thus, the presence or absence of a correlation of fluorescence intensity will be difficult to analyze. For this reason, we did not perform such a correlation.

(4) Figure 1c shows some, but not convincing co-localization of the autoantibodies with Homer-1c. Co-localization with a commercial or commonly accepted GluN1 antibody would more directly demonstrate the specificity of the autoantibodies.

Reply: We have performed co-immunostaining of purified IgG from Healthy-, Healthy+ and PSY+ samples with a commercial anti-GluN1 antibody targeting the extracellular part of the GluN1 subunit. We can clearly observe a co-localization between the Healthy+ and PSY+ autoantibodies staining and the endogenous NMDAR labelling (new Fig. 1d). On the opposite, the labelling with purified IgG from the Healthy- sample shows weaker fluorescence intensity and likely does not co-localize with NMDAR. We confirmed these results using ectopic expression of GluN1-SEP both in dissociated hippocampal neurons and in HEK cells (Fig. 1c). These new data have now been added to the revised manuscript.

(5) The authors should consider to re-do the absorption assay as in Figure 3g, and test whether staining with the autoantibodies is lost after pre-absorption on GluN1/2 transfected HEK cells (also see below).

Reply: We did perform immunostaining with purified human IgG of HEK cells before and after absorption assay. As shown in the images below, the absorption assay reduced the human IgG staining on transfected HEK cells, as previously published (Graus et al., Neurology, 2008). This point and reference has now been added in the revised manuscript.

NMDAR-Ab staining relative to the absorption assay.

Upper panel: Control condition where residual background is observed after incubation (>12h) of purified IgG from a seropositive sample (human IgG) on non-transfected HEK cells. *Lower panel:* co-immunostaining of GluN1-SEP expressing HEK cells with purified IgG from a seropositive sample pre- and post-absorption of NMDAR-Ab. Note that the colocalized signals between GluN1-SEP positive HEK cells (green) and human IgG staining (red) is strongly reduced after absorption.

(6) Ideally, the authors would provide data that IgG staining is absent in tissue or neurons from GluN1 knockout mice, or reduced in GluN1 knockdown neurons.

Reply: We do agree with the reviewer that performing IgG staining on tissues where GluN1 expression is genetically abolished would be a robust control. However, we do not have such GluN1 knock-out mouse to our disposal. Still, we did perform such test in HEK cells that intrinsically do not express NMDAR and we did not observe any staining on non-transfected cells, confirming that the NMDAR is the target of autoantibodies.

The experiments presented in Figure 3 lack the comparison with a proper negative control, e.g. untreated cells or cells treated with an unrelated human IgG. This is in fact also the case for experiments presented in Figure 4 and 6, see below. The authors should include this, without this control it is difficult to interpret the differential effect of the Healthy+ and PSY+ derived autoantibodies on NMDAR diffusion.

Reply: Indeed, we initially performed our experiments with purified IgG from a healthy negative subject and a commercial anti-GluN1 antibody (AGC-001, Alomone Labs) targeting an extracellular epitope of the GluN1 subunit (amino acid residues 385-399). For a purpose of clarity, we decided to only show the comparison between Healthy+ and PSY+ NMDAR-Ab conditions. We somehow agree with the reviewer that the lack of control conditions in the graphs may be confusing. We have thus included these data in the revised manuscript. We observed that synaptic NMDAR diffusion is similarly increased after incubation with a control α GluN1_{N-term} IgG or Healthy+ NMDAR-Ab compared to Healthy- IgG, suggesting that the presence of an antibody targeting the extracellular part of the NMDAR slightly perturb its trafficking properties when examined at the single molecule level (no change at the macroscopic level) (Fig. 3c). IgG binding to extracellular epitopes may indeed change the receptor's environment, possibly due to steric hindrance. Strikingly, PSY+ NMDAR-Ab show a strong, nano- and macroscopic impact on synaptic NMDAR dynamic distribution compared to the control α GluN1_{N-term} IgG or Healthy+ NMDAR-Ab, strengthening a disturbing role. As noted, neither Healthy+ NMDAR-Ab nor none of the control conditions (no IgG, α GluN1_{N-term} or Healthy-IgG) had any macroscopic impact (Supplementary Fig. 5c) but only PSY+ NMDAR-Ab significantly impaired NMDAR synaptic content (Fig. 5b).

The experiment presented in Figure 3g is a very important and elegant approach to test the specificity of the autoantibodies (but on itself not sufficient). However, the authors describe that they do the pre-absorption on GluN1-transfected cells, while they describe earlier that these autoantibodies did not stain HEK cells that only express GluN1, suggesting that these do not recognize GluN1 in isolation, only in the presence of GluN2 subunits. The authors should address this contradiction.

Reply: We apologize for the lack of clarity; the legend of Figure 3g has been corrected to avoid any misunderstanding. HEK cells were co-transfected with both GluN1 and GluN2B subunits to allow a proper surface expression of ectopic NMDAR. We then performed live immunostaining to ensure surface NMDAR labelling. However, we do observe IgG staining when GluN1 is expressed alone in HEK cells, suggesting that the presence of GluN2 subunits is not necessary for autoantibodies binding.

Figure 4 presents super-resolution images of GluN2A-containing NMDARs and reveals 'nano-objects' that are clustered together in groups of 4-6. This is a very intriguing finding, and as the authors note, one of the first investigations into the nanoscale organization of NMDARs. However, data for untreated control cells is lacking entirely. This should be included for comparison with the experimental conditions. Perhaps it would also be interesting to compare measures of domain organization between extrasynaptic and synaptic clusters. Also, could the authors elaborate a little bit more on the function of these domains? Perhaps the authors could discuss this finding in the light of other comparable studies describing these domains for other synaptic proteins (e.g. Nair et al., 2013, MacGillivray et al., 2013, Specht et al., 2013).

Reply: To the best of our knowledge, the nanoscale distribution of surface NMDAR has not been reported. In another parallel study (in the process of publication), we have performed a full description of NMDAR subtype distributions using superresolution imaging in hippocampal cultured neurons. To directly answer the reviewer's comment, we find that the average number and area of endogenous GluN2-NMDAR nano-domains onto naïve hippocampal neurons are similar to the ones described in this report, indicating that the presence of IgG does not alter the nano-domains number and area. Regarding the function of these nano-domains, one can indeed speculate that these structures represent the core apparatus of the synaptic NMDAR-mediated transmission. Our yet unpublished data further demonstrate that GluN2A- and GluN2B-NMDARs are differently regulated at the nanoscale level, indicating that this organization is functionally relevant and implicate specific cellular cascades. In the revised manuscript, we have now inserted these comments as suggested by the reviewer (e.g. page 15).

Figure 6 presents data that EphB2R levels are reduced by autoantibodies from psychotic patients. However, also in neurons treated with autoantibodies from the healthy group, only 40% of the synapses are EphB2R positive. Is this a normal fraction found also in control neurons, or do the antibodies from the healthy+ group also affect EphB2R levels? The authors should present data for a proper negative control group.

Reply: Our data are in line with previous electron microscopy observations (e.g. Grunwald et al., Nat. Neurosci., 2004), reporting that EphB2 receptors are found in about 40% of CA1 synapses. In the revised manuscript, we have added the control condition with purified IgG from a seronegative subject (see Fig. 6a-b), further supporting our initial claim.

The authors claim that "...our data demonstrate that NMDAR-Ab from PSY+ patients specifically disturb EphB2R synaptic retention and content, suggesting that the loss of this anchoring partner of NMDAR is likely responsible for the destabilization of the receptor in presence of PSY+ NMDAR-Ab." The evidence for their conclusion that the autoantibodies disrupt the interaction between synaptic EphrinB2 receptors and thereby uncouple NMDARs, however is very weak. The only piece of evidence that is provided is that the synaptic levels of both NMDARs and EphB2Rs are reduced after chronic exposure to the autoantibodies. This is in line with their previous work, but on itself not more than an interesting correlation. The authors should provide more compelling evidence that the autoantibodies disrupt the interaction between EphB2R and NMDARs, or adjust their conclusion.

Reply: We indeed previously showed, using co-immunoprecipitation experiment, that encephalitis NMDAR-Ab reduce the interaction between EphB2 receptor and GluN1 subunit (Mikasova et al., Brain, 2012). It is worth mentioning that among the various experimental approaches used, the strongest alteration was observed using single molecule imaging and surface immunostaining, as they specifically isolate surface receptors and are thus more sensitive to the action of extracellular NMDAR-Ab. In order to address the point of the reviewer, we did perform several series of co-immunoprecipitation experiments between EphB2 receptor and GluN1 subunit in cultured neurons exposed to either Healthy+ or PSY+ NMDAR-Ab. The relative NMDAR-EphB2R interaction was not different between the two conditions (*Healthy+ = 1.000 ± 0.06661, n = 10 cultures; PSY+ = 1.006 ± 0.06285, n = 13*). The lack of co-IP effect may appear, at first glance, surprising. However, we show in Figure 1 that the titers of NMDAR-Ab from psychiatric patients are 8 times lower than the ones from encephalitis patients. As encephalitis NMDAR-Ab strongly impact the membrane interaction (e.g. single receptor tracking) but only slightly alter the overall interaction (co-IP) (see Mikasova et al., 2012), we were not that surprised to see that the titer of PSY+ NMDAR-Ab is likely not sufficient to display a "biochemical effect". Thus, in light of the reviewer's comment and this new dataset, we have tuned down our conclusion that PSY+ NMDAR-Ab may mechanistically act through the disruption of the NMDAR-EphB2 receptor, a hypothesis that needs to be thoroughly investigated in the future using sensitive methods to target surface receptor interaction.

Also, the conclusion that this is specific for EphB2R is too far of a stretch, as they for instance also found a reduction in AMPAR levels, seemingly indicating a more general effect on synaptic receptor levels. In that respect, alternative explanations are hardly explored or discussed. For instance, perhaps the most straight-forward explanation for reduced surface expression is that IgG binding to NMDARs induces their internalization (along with interactors such as EphB2R). This should be addressed experimentally, or at least be discussed as an alternative explanation for their results.

Reply: Our initial intent was not to state that the impact of NMDAR-Ab on NMDAR trafficking is solely dependent on the EphB2R interaction. We believe, as the reviewer, that the interaction is part of a cascade in which NMDAR-Ab trigger a synaptic destabilization of NMDAR. In line with the above answer, we have edited the manuscript in order to avoid such a misunderstanding.

The authors present their single-molecule tracking approach as an unambiguous assay to test the specificity of the human IgGs. Even further, in the discussion the authors propose that their single-molecule approach could be a diagnostic standard to detect NMDAR-Abs in human serum. They do point out that this is technically challenging, but besides this and other practical considerations (e.g. experimental reproducibility), at best, this approach will only provide evidence that the autoantibody binds an epitope on a protein that has similar mobility characteristics as the NMDAR: it cannot provide unambiguous evidence that the IgG at test is an NMDAR-specific Ab. The authors should re-phrase or remove this statement from the manuscript.

Reply: This point has been rephrased in the manuscript. Our intent was to state that single molecule, in addition to classical assays (daily used in all hospitals around the world), demonstrates the presence of NMDAR-Ab. The single molecule assay by itself is surely insufficient, as it only provides evidence that a membrane target with similar behavior as the NMDAR is detected by the autoantibodies.

The authors conclude that "...only patients' NMDAR-Ab prevent chemically-induced AMPA receptor synaptic potentiation while leaving intact NMDAR-mediated calcium influx". However, these two outcomes (intact calcium influx, but no potentiation) are measured at completely different time scales, after 5 minute vs. 12 hr incubation times. After 12 hr incubation with the PSY+ autoantibodies NMDAR levels were severely reduced, and it is very likely that at that point NMDA-mediated calcium influx is also disrupted. Thus, it is unknown whether the effect on plasticity is independent of the effect on NMDAR-dependent calcium signaling. These two conclusions should be more clearly separated, or it should be tested experimentally that calcium influx is not disrupted after 12 hr incubation with IgGs.

Reply: We agree that this statement generated confusion and we apologize for that. Indeed, after 12h incubation with NMDAR-Ab the NMDAR-dependent calcium signaling is more than likely altered as we provide evidence that the NMDAR synaptic content is strongly altered. We have now rephrased this sentence to indicate that the effect observed on LTP is likely not due to a "classical" antagonist effect of NMDAR-Ab.

To me the determination of the autoantibody titers is confusing. It is described that "titers were estimated as end point dilutions". Could the authors explain this in a little bit more detail? What is the unit in Figure 1b? The term 'CBA' is also used in the discussion (line 330), but not explained.

Reply: Due to limited volumes of serum for each patient, we were not in capacity of extracting pure fractions of NMDAR-Ab IgG. To overcome this limitation and assess the levels of NMDAR-Ab present in purified IgG fractions, we estimated NMDAR-Ab titers using end-point dilutions. Practically, GluN1/GluN2B transfected HEK cells were incubated with serial dilutions of patient's serum. The dilution which corresponds to a loss of staining provides an estimation of the NMDAR-Ab titer thus called "end-point dilution". The unit of Figure 1b is then a dilution. The term CBA, i.e. cell-based assay, was detailed in the method section and is now better defined in the new version of the manuscript.

In Figure 8d, the authors present a quantification of GluA1 cluster area. Presenting intensity data would be more appropriate here as an increase in AMPAR levels is expected, not necessarily synaptic area. Also, with respect to the statistical analysis, it seems that in 8b and 8d the IgG groups are each compared with the control group. Is the PSY+ group also significantly different from the Healthy+ group?

Reply: In Figure 8, we indeed expressed two variables of the GluA1-AMPA surface expression. To examine the long-term effect of NMDAR-Ab (>12h incubation), we performed an immunostaining of GFP on GluA1-SEP expressing neurons. With this protocol, the GluA1-SEP signal was amplified (anti-GFP primary antibody + secondary antibody coupled to an Alexa) and the fluorescence signal was high and stable between cells allowing us to analyze quantitatively variations between conditions. To assess the effect of a cLTP stimulus in presence of NMDAR-Ab, we performed live experiments in which GluA1-SEP clusters were monitored before and after cLTP. Practically speaking, GluA1-SEP transfected neurons were incubated for >12h with NMDAR-Ab

and synaptic AMPAR content was measured before and after a cLTP stimulus. The quantification of this live signal is however more challenging (than the one from fixed immunostained AMPAR) as it fluctuates over time and is surely weaker in fluorescence intensity. Based on our experience, the fluorescence intensity of GluA1-SEP clusters is subjected to higher stochastic variations than the area of the clusters, explaining why we mostly use the cluster area in this case. To address the point of the reviewer, we have now included the distribution of GluA1-SEP fluorescence intensity clusters as a Supplemental material. These data show that without any NMDAR-Ab the intensity of live GluA1-AMPA is significantly increased after a cLTP stimulus. In presence of Healthy+ NMDAR-Ab, the distribution shows that a sub-population of synapses is strongly potentiated (the one with a lower basal AMPAR content). Importantly, in presence of PSY+ NMDAR-Ab, we no more observed any potentiation of GluA1-AMPA clusters. Thus, these data complement the robust “cluster area” analysis and indicate that PSY+ NMDAR-Ab prevent cLTP-induced GluA1-AMPA potentiation. Noteworthy, these *in vitro* experiments have now been reproduced *in vivo/ex vivo* in the revised manuscript.

Reviewer #2

We thank the reviewer for her/his positive and constructive comments, which strengthen our manuscript.

It is very much unclear to me when the authors used pooled human antibodies and when they tested each sample independently. This is of critical importance for the conclusions the authors are making. Do every sample from patients cause the synaptic NMDAR disruption and does every sample from controls not cause the disruption? If pooled, the effects could be induced by a single dominant sample, and the ability to make any sort of generalization is lost. If individual samples were used, the presented data in each figure needs to represent the variation between samples rather than, for example, variation between regions of interest in the imaging. This is a crucial distinction.

Reply: We apologize for the lack of clarity. The revised manuscript has been edited accordingly. Each purified IgG sample has been tested individually (4 different PSY+ patients and 3 healthy+ subjects) but to clarify the message, data were pooled per experimental condition after controlling that the variability within each clinical condition allowed us to group them. Indeed, we did not observe any significant difference between PSY+ patients (see Supplementary Fig. 3) nor between Healthy+ individuals (see below). The number of samples used in each set of experiment now appears in the revised manuscript.

Intra-variability within the Healthy+ group. Cumulative distributions of instantaneous diffusion coefficient of synaptic GluN2A-NMDAR exposed to control IgG ($\alpha\text{GluN1N-term}$) or Healthy+ NMDAR-Ab. $\alpha\text{GluN1N-term}$ = $0.0651\mu\text{m}^2/\text{s}$, IQR = $0.0186 \pm 0.1502 \mu\text{m}^2/\text{s}$, $n = 3475$ trajectories. Patient 1 = $0.0615 \mu\text{m}^2/\text{s}$, IQR = $0.0227 \pm 0.1541 \mu\text{m}^2/\text{s}$, $n = 473$ trajectories. Patient 2 = $0.0929 \mu\text{m}^2/\text{s}$, IQR = $0.0251 \pm 0.1777 \mu\text{m}^2/\text{s}$, $n = 592$ trajectories. Patient 3 = $0.0685 \mu\text{m}^2/\text{s}$, IQR = $0.0217 \pm 0.1547 \mu\text{m}^2/\text{s}$, $n = 592$ trajectories.

It was noted that in no cases were anti-NMDAR antibodies detected in the patients' CSF. If circulating CSF antibodies are not found in the brain, this limits the clinical utility of these findings. The authors, however, tested for CSF antibodies on fixed HEK cells whereas the earlier said with serum samples that fixed HEK cells did not work for detection and they thus used live HEK cells. Why the difference? Also, how does the concentration of Ab used in these experiments relate to the Ab exposure of synaptic NMDARs, especially if there is none in the CSF?

Reply: The absence of NMDAR-Ab in the CSF of psychiatric patients raises the question of the clinical utility. This question is part of a large and intense debate in the field as the presence of autoantibodies in the serum, but not in the CSF, of patients with neurological and psychiatric conditions has been constantly reported. The main hypothesis is that the blood-brain barrier can be permanently or temporarily breached in neuropsychiatric disorders, as it is often associated with inflammation. The circulating autoantibodies would thus penetrate the brain and act on their targets. The concentration of circulating NMDAR-Ab used in this study could thus theoretically be related to the pathology. Other scenario, such as an initial peripheral effect of the autoantibodies that will trigger brain dysfunction in a secondary process, are possible, though less supported by fundamental and clinical observations. This point is briefly discussed in the manuscript.

Regarding the different methods used to detect NMDAR-Ab, the rationale is based on the well-accepted observation that CSF produces much less background staining than serum. Thus, most laboratories use fixed HEK cells to detect the presence of autoantibodies in the CSF, as the signal-to-noise ratio of this test is excellent. With serum, the background staining in fixed HEK cells is too high, generating lower signal-to-noise ratio.

For many figures it would be useful to have additional controls. Figure 1 – positive controls with commercial antibodies, negative controls of purified IgG from negative subjects, negative controls from untransfected HEK cells, etc. For other experiments, purified human IgG from negative subjects as a negative control would be useful. The distinction between the schizophrenia+ samples and the healthy+ samples is very interesting, but hard to know if the healthy+ samples are doing anything.

Reply: As mentioned previously, we decided to only show Healthy+ versus PSY+ conditions in the initial manuscript for a purpose of clarity and not to dilute the message of the study. However, we do understand the concern of the reviewer and control conditions appear in the new version of the manuscript. Staining with positive and negative controls have been added to Figure 1 and results obtained with purified IgG from a seronegative sample have been added to other experiments.

The reduction of EphB2R from synapses in Figure 6 is interesting, does this suggest a disruption of the interaction with NMDARs or perhaps they are moving together? Ideally, some evidence of a disruption of the interaction would be supportive, though at the minimum, addition discussion is warranted.

Reply: Our data indeed support the idea that PSY+ NMDAR-Ab alter the interaction between the EphB2 receptor and the GluN1 subunit. This is reminiscent from our previous work with encephalitis NMDAR-Ab in which we showed that encephalitis NMDAR-Ab reduce the GluN1- EphB2R interaction (Mikasova et al., Brain, 2012). In an effort to directly show that the interaction is indeed disrupted by the NMDAR-Ab, we performed a series of co-immunoprecipitation experiments between the EphB2 receptor and the GluN1 subunit in cultured neurons exposed to either PSY+ or healthy+ NMDAR-Ab. The relative interaction between EphB2 receptor and GluN1 subunit was not different between the two conditions (*Healthy+ = 1.000 ± 0.06661, n = 10 cultures; PSY+ = 1.006 ± 0.06285, n = 13*). The lack of effect on the co-immunoprecipitation may appear, at first glance, surprising. However, the titers of NMDAR-Ab from psychiatric patients are 8 times lower than the one from encephalitis patients. As encephalitis NMDAR-Ab strongly impact the membrane interaction (e.g. single receptor tracking) but only slightly alter the overall interaction (co-immunoprecipitation) (see Mikasova et al., 2012), we were not that surprised to see that the titer of PSY+ NMDAR-Ab were not “sufficient” to provide a clear biochemical effect in our preparation. For future projects, we plan to develop FRET/BRET imaging to investigate the interaction between these receptors at the plasma membrane with high sensitivity. Thus, we have adjusted our conclusion based on the reviewer’s comment, tuning down our interpretation and suggesting additional experiments beyond the scope of this study.

Figure 7 does not provide enough evidence to make the conclusion that anti-NMDAR antibodies do not effect Ca²⁺ transients. The frequency of the events says very little about the NMDAR responses if there is not a loss of synapses or complete loss of NMDARs at synapses. Analysis of the amplitude, area, and decay kinetics of the Ca²⁺ transients is needed at the minimum. Electrophysiological analysis in these samples would be additionally be welcome.

Reply: In Figure 7, we used the high signal-to-noise Ca²⁺ probe GCaMP3, which provides a fast and very good sensitivity to calcium changes, to study the impact of NMDAR-Ab on NMDAR-dependent activity. We showed that neither purified IgG from Healthy- subjects nor NMDAR-Ab from Healthy+ or PSY+ individuals affect the frequency of NMDAR-dependent Ca²⁺ events, supporting the idea that NMDAR-Ab do not acutely modulate NMDAR-dependent activity. We agree with the reviewer that the frequency of Ca²⁺ transients may not provide a full evaluation of NMDAR-Ab effects on NMDAR-dependent activity. We thus followed the reviewer's recommendations and performed a deeper examination of the Ca²⁺ signal dynamics. Further analysis did not reveal any difference in the area (under the curve), rise or decay times of NMDAR responses after exposure to Healthy+ or PSY+ NMDAR-Ab. These new elements are now included in the text of the revised manuscript. Nevertheless, we concede that an electrophysiological analysis (e.g. NMDAR single-channel recordings in heterologous systems) would provide critical information regarding the effect of NMDAR-Ab on NMDAR channel properties. This question albeit very exciting represents a technically ambitious research project as the amount of material available from patients (μ l of NMDAR-Ab) make this classical electrophysiological experiments difficult to realize. Instead, we decide to use our precious little amount of human material to fully cover the next point of the reviewer.

Figure 8 would likewise be more supported with some electrophysiological analysis.

Reply: To address this comment, we performed a series of intra-hippocampal injections of NMDAR-Ab in young rats (P12-P15), followed by patch-clamp recordings at CA3-CA1 synapses. Long-term potentiation (LTP) of these synapses was induced by a high frequency stimulation protocol and excitatory postsynaptic currents (EPSC) were recorded at -60mV. This new dataset shows that NMDAR-Ab from PSY+ subjects prevent LTP, even inducing a long-term depression, at these synapses. Together, our *in vitro* and *ex vivo* recordings strengthen the same conclusion that NMDAR-Ab from PSY+ patients specifically alter LTP at hippocampal synapses. This important new set of data has been incorporated into the revised manuscript.

Are these effects GluN2A-containing receptor specific, or are there also effects of GluN2B-containing receptors?

Reply: NMDAR-Ab apparently target the extracellular part of the GluN1 subunit, so one could hypothesize that both GluN2A and GluN2B subunits are affected in a similar way. However, the relative contribution of GluN2A and GluN2B diheteromers, and GluN2A/GluN2B triheteromers to the NMDAR synaptic pool is still debated. Thus, in our study we fully concentrated on the impact of NMDAR-Ab on the GluN2A-NMDAR as it is well-accepted that this subunit is highly enriched in synapses. To please the reviewer, we performed a series of experiments in which we measured the overall dynamics of GluN2B-NMDAR after 30min incubation with NMDAR-Ab from one psychotic patient. We report that PSY+ NMDAR-Ab increase GluN2B-NMDAR diffusion compared to Healthy+ NMDAR-Ab (*GluN2B-NMDAR median instantaneous diffusion coefficient \pm 25-75% interquartile range in presence of Healthy+ NMDAR-Ab = 0.0251 μ m²/s, IQR = 0.0046-0.0953 μ m²/s, n = 5261 trajectories, or PSY+ NMDAR-Ab = 0.0317 μ m²/s, IQR = 0.0054-0.0978 μ m²/s, n = 4113 trajectories; **p = 0.0035, Mann-Whitney test). This supports the view that the GluN1 subunit is the target of the antibody. However, a full analysis of the effect of NMDAR-Ab from each patient and in different membrane compartments (synaptic, perisynaptic and extrasynaptic areas) is warranted to claim that both GluN2A- and GluN2B-NMDAR are equally affected by NMDAR-Ab. Due to limited amount of material, we consider that this additional information is beyond the reasonable scope of this study.*

Why is the MSD in figure 1f so different from the MSD figure 3d?

Reply: We apologize for this mistake. The scale of the MSD graph in Figure 1f has been corrected in the new version of the manuscript.

Minor issues:

- Line 37 (abstract): should "liability" be "reliability"?
- Line 62: associated with psychotic symptoms
- Line 117: maybe "past" instead of "ancient"?
- Line 121: I think this refers to supplementary table 1 instead of 3
- Unsure why authors used "PSY+" instead of "SCZ+" to be consistent throughout manuscript
- Suppl Figure 1 could use some sort of quantification to say "comparative labeling was observed" on line 155.
- Line 167: please provide a reference to support this statement
- Figure 2e either needs much-improved labeling and/or should be moved to supplemental data
- Suppl Figure 3b: I don't see the commercial antibody line (black line in the key) on this graph
- The pre-absorption experiment was clever... not an issue, just wanted to say I liked it
- Suppl Figure 4b: how do you explain the reduction in synaptic GluA1 in Figure 8a without any changes in GluA1 movement? Perhaps add discussion. The most straightforward explanation is that NMDAR-Ab do not alter acutely (tens of minutes) the dynamic equilibrium of surface AMPAR. However, a chronic exposure (>12h) to NMDAR-Ab alter synaptic NMDAR content, as well as the AMPAR content. The long-term effect is likely a consequence of the altered NMDAR synaptic signaling.

These typos have been corrected.

- Figure 4a and Line 244: how do you know the surface GluN2A is "in glutamatergic synapses"?

We acknowledge that this wording may be misleading and the sentence has been rephrased in the new version of the manuscript. Surface GluN2A-NMDAR are undoubtedly present in and out of glutamatergic synapses. Here, PSD-95 was used as a synaptic marker in order to study GluN2A synaptic localization specifically.

- Lines 248-250: Sentences for Fig 4c and 4d seem swapped and should be in order of figure.
- Line 290: I don't like the use of "per se" here
- Line 312-313: "impair the plastic range of glutamatergic synapses" is overreaching.

All these points have been addressed in the new version of the manuscript.

Reviewer #3

In this study, Jézéquel et al reported that autoantibodies against NMDAR (NMDAR-Ab) is differentially expressed in some portion of patients with schizophrenia (SZ) and healthy controls. The authors also demonstrated that the surface dynamics of synaptic NMDAR and EphrinB2, its anchoring receptor, is altered by treatment with NMDAR-Ab collected from patients with SZ, but not those from healthy controls using cutting-edge technologies, including single molecules imaging with Quantum dots and super-resolution microscopy. Quantum dot method was the technology that the authors previously used. In fact the authors previously demonstrated that NMDAR-Ab from patients with cognitive deficits prevent synaptic potentiation without affecting NMDAR function (Dupuis et al, EMBO Journal 2014). Thus the methods have been used by this group before and are not novel (though still cutting-edge). The results of the effect of NMDAR-Ab from patients with SZ are interesting, however, the sample size is too limited (3 seropositive healthy control and 4 seropositive SZ patients [though 9 SZ patients were identified as seropositive, only 4 is available for experiments]) to obtain definitive assessment. This reviewer is not convinced from the current data to see the significance to schizophrenia.

Reply: The reviewer has two criticisms about our work: i) the methods and ii) the sample size. Regarding our experimental approaches, it is stated that “the methods have been used by this group before and are not novel”. We completely agree that we did use, and publish work with, single molecule imaging, classical confocal imaging, superresolution imaging (STORM), calcium imaging, and electrophysiology. Our study did not intend to publish an innovative new technology for brain imaging but rather to dissect some of the molecular disturbances induced by NMDAR-Ab. There is thus here a clear misunderstanding. We have thus edited the revised manuscript in order to avoid such a misinterpretation.

Regarding the sample size, we fully agree that a higher number of seropositive healthy individuals and patients is always better. However, establishing and thoroughly exploiting a cohort of healthy individuals and patients, as we did here with respect to the neurological and psychiatric history, is not trivial and would likely require the establishment of an international consortium. Most importantly, we found that the differential effect of NMDAR-Ab is consistent and reproducible between the two groups, even with a limited number of individuals, which in our view strongly strengthens our conclusion. Indeed, NMDAR-Ab from each patient had the potency to displace NMDAR-Ab.

Thus, we believe that our report sheds lights on the molecular disturbance induced by NMDAR-Ab from psychotic patients. Whether our insights are, or are not as stated by the reviewer, of “significance for schizophrenia” remains surely an open question for the field that will be solved within the next decades. This exact same question is central in our field, as for instance the real clinical impact of major discoveries reporting genetic, epigenetic, immune, or other alterations in schizophrenic patients (e.g. Won et al., Nature, 2016; Schizophrenia Working Group of the Psychiatric Genomics Consortium, Nature, 2014; Purcell et al., Nature, 2014; ...). We do agree that psychotic disorders are surely complex, potentially heterogeneous, in their origins. In the revised manuscript, we made sure that we do not convey the conclusion that we “solved” the origin of schizophrenia.

Reviewer #4

We thank the reviewer for her/his positive and constructive comments, which strengthen our manuscript.

To further demonstrate this point, it would be critical to show that the autoantibodies from patients and controls are binding to different epitopes on the NMDAR. In the absence of these data, it the authors should at least discuss the epitopic differences as an explanation for the differential autoantibody function between psychosis positive and negative antibody positive subjects. Further, if one wanted to design a diagnostic, as the authors propose, it would be best to use the epitope peptides as they would likely be pathologically significant based on the evidence presented in this manuscript.

Reply: The question of the epitope is indeed crucial. The N368 residue located on the extracellular part of the GluN1 subunit of the NMDAR was proposed as a necessary component for the epitope formation of NMDAR-Ab in NMDAR-Ab encephalitis (Gleichman et al., J. Neurosci., 2012). Possibly due to the restricted amount of psychiatric patients with NMDAR-Ab and to the tedious technique of epitope mapping, only one study explored NMDAR-Ab binding in schizophrenia (SCZ) and other clinical conditions and reported no change of NMDAR-Ab binding from SCZ patients after mutation of the N368 region (Castillo-Gómez et al., Mol. Psychiatry, 2016). Using a classical immuno-competition assay, we here tested whether NMDAR-Ab from different clinical conditions (healthy, SCZ and NMDAR-Ab encephalitis subjects) share (a) common epitope(s). Our results suggest the existence of several binding sites and support the idea that NMDAR-Ab do not share a unique epitope, even within the same group of individuals. Hence, the search for NMDAR-Ab epitopes in SCZ represents a major objective to explore inter-individuals differences and elucidate the mechanisms underlying antibody pathogenicity.

Reviewers' comments:

Reviewer #1 (Remarks to the Author):

The authors have addressed most of my concerns, and the manuscript has been improved significantly. In particular, the addition of control groups and the electrophysiological experiments showing a defect in the expression of LTP make the manuscript much stronger, and support the proposed idea that autoantibodies from psychotic, but not from healthy subjects affect NMDAR diffusion and function.

- however, the data does still not completely convince that the autoantibodies exclusively recognize NMDARs. The now comprehensive labeling studies on HEK cells convincingly show that these autoantibodies can recognize the expressed NMDAR subunits, and do not recognize a few other synaptic surface proteins. However, the distribution patterns in neurons (Figure 1c and d) for the human IgGs show only very little co-localization with SEP-GluN1 (1c), or endogenous GluN1 (1d). In these examples, human IgGs recognize some, but certainly not all endogenous GluN1 puncta, and there are many IgG-labeled patches on the dendrite that are not labeled by anti-GluN1 or SEP-GluN1. Yet, the authors describe these results as "clearly overlapped" distribution patterns. This point should be addressed more thoroughly. I understand that it is not feasible to perform a control staining on GluN1 KO animals, but effective GluN1 knockdown constructs have been published (e.g. Alvarez et al., J Neurosci 2007), and a reduction in staining on GluN1 knockdown neurons would provide stronger support for the claim that these autoantibodies recognize endogenous NMDARs in hippocampal neurons.

- in figure 1d, the image quality for the Healthy+ example (middle panel) is poorer compared to the other examples, and it seems that the left panel in this row is differently cropped than the middle panel and the overlay: puncta at the bottom that are visible in this panel are not visible in the overlay. Same for panels in Figure 1a, the overlays in the middle and bottom panels are shifted with respect to the middle panel.

- In figure 6c, the y-axis displays a fraction, not percentage. The label or numbering should be adjusted.

- The experiments in Figure 6d and 6e still lack a Healthy- control, this control group should be included.

- The notion that also control anti-GluN1 IgGs alter the diffusion properties of NMDARs is noteworthy. This control group is now included in Figure 3, but it is not described in the text. This should be added.

Reviewer #2 (Remarks to the Author):

All major issues raised have been appropriately addressed and provides for a clearer and

better-controlled study.

Reviewer #3 (Remarks to the Author):

The authors have sufficiently addressed all concerns from this reviewer.

Reviewer #4 (Remarks to the Author):

The authors adequately addressed my issues.

We first would like to thank the reviewers for their positive feedbacks, emphasizing our strong efforts to address all their critics. We have now performed additional experiments to address the additional critics of reviewer 1.

Reviewer #1

The data does still not completely convince that the autoantibodies exclusively recognize NMDARs. The now comprehensive labeling studies on HEK cells convincingly show that these autoantibodies can recognize the expressed NMDAR subunits, and do not recognize a few other synaptic surface proteins. However, the distribution patterns in neurons (Figure 1c and d) for the human IgGs show only very little co-localization with SEP-GluN1 (1c), or endogenous GluN1 (1d). In these examples, human IgGs recognize some, but certainly not all endogenous GluN1 puncta, and there are many IgG-labeled patches on the dendrite that are not labeled by anti-GluN1 or SEP-GluN1. Yet, the authors describe these results as “clearly overlapped” distribution patterns. This point should be addressed more thoroughly. I understand that it is not feasible to perform a control staining on GluN1 KO animals, but effective GluN1 knockdown constructs have been published (e.g. Alvarez et al., J Neurosci 2007), and a reduction in staining on GluN1 knockdown neurons would provide stronger support for the claim that these autoantibodies recognize endogenous NMDARs in hippocampal neurons. authors present several tests to demonstrate the ability of autoantibodies to detect NMDARs, but it is unclear how specific these autoantibodies are for NMDARs. Especially the images in Figure 1d are very hard to interpret, there is clear staining with the human IgGs visible in GFP-negative areas. Also, the other way around, in the PSY+ group, a clear GFP-positive area is IgG-negative. The degree of specificity of the autoantibodies for the NMDAR has important implications for the interpretation of the results, and the authors should address this by doing at least some of the following controls: (1) The authors should include (images of) appropriate negative controls: human IgG staining on untransfected HEK cells, and/or human IgG staining on HEK cells transfected with another GFP-tagged synaptic surface protein.

Reply: The reviewer expresses doubts about the fact that NMDAR-Ab are indeed directed against NMDAR. Although the approaches used in this manuscript have been validated for the detection of NMDAR-Ab in former studies, we have now imported and processed brain tissues from GluN1 knockdown mice to directly address the reviewer's critic. Schematically, brain sections from wild-type and GluN1 KD mice were exposed to PSY+ IgG and secondary staining with fluorescent anti-human IgG was processed. As shown in the new Figure 1 (panel e), clear staining of the hippocampal CA1 area was obtained in the WT mice. Expectedly, in GluN1 KD tissue there was virtually no staining observed, just low background fluorescence. Thus, these observations further strengthen our conclusion that anti-NMDAR IgG bind GluN1 subunit, with no evidence of the presence of additional IgG against other brain targets. This dataset has now been included into the revised manuscript.

- in figure 1d, the image quality for the Healthy+ example (middle panel) is poorer compared to the other examples, and it seems that the left panel in this row is differently cropped than the middle panel and the overlay: puncta at the bottom that are visible in this panel are not visible in the overlay. Same for panels in Figure 1a, the overlays in the middle and bottom panels are shifted with respect to the middle panel.

Reply: We took the reviewer's comments into account and corrected the corresponding panels from figures 1a and 1d accordingly. These are included in the revised manuscript.

- In figure 6c, the y-axis displays a fraction, not percentage. The label or numbering should be adjusted.

Reply: This has been corrected.

- The experiments in Figure 6d and 6e still lack a Healthy- control, this control group should be included.

Reply: For purely technical reasons we were not able to deliver such specific additional dataset. Indeed, we simply do not have sufficient amount of Healthy- material to run such a live experiment. As this cohort material was very precious we prioritized the experiments, and this additional control

was not in our highest priority. Indeed, we would like to mention that the healthy- group was performed for the immunostaining of the EphB2R, strengthening our conclusion. Although we agree

on the principle that completing this with single molecule tracking will be optimal, we simply cannot perform this experiment.

- The notion that also control anti-GluN1 IgGs alter the diffusion properties of NMDARs is noteworthy. This control group is now included in Figure 3, but it is not described in the text. This should be added.

Reply: These data have now been discussed in the result section (page 9, second paragraph).

Reviewer #2

Satisfied with our previous additional set of data.

Reviewer #3

Satisfied with our previous additional set of data.

Reviewer #4

Satisfied with our previous additional set of data.

REVIEWERS' COMMENTS:

Reviewer #1 (Remarks to the Author):

All my major issues have been adequately addressed. The provided data showing absence of staining in the GluN1 knockout mouse make it a very convincing, well-controlled study.

We first would like to thank the reviewers for their positive feedbacks.

Reviewer #1

Reviewer #1 (Remarks to the Author):

All my major issues have been adequately addressed. The provided data showing absence of staining in the GluN1 knockout mouse make it a very convincing, well-controlled study.

Response: We thank the reviewer for his/her comment.